# Density Ratio Estimation with Conditional Probability Paths

**Hanlin Yu** [1]  **Arto Klami** [1]  **Aapo Hyvärinen** [1]  **Anna Korba** [2]  **Omar Chehab** [2]

## Abstract

Density ratio estimation in high dimensions can be reframed as integrating a certain quantity, the time score, over probability paths which interpolate between the two densities. In practice, the time score has to be estimated based on samples from the two densities. However, existing methods for this problem remain computationally expensive and can yield inaccurate estimates. Inspired by recent advances in generative modeling, we introduce a novel framework for time score estimation, based on a conditioning variable. Choosing the conditioning variable judiciously enables a closed-form objective function. We demonstrate that, compared to previous approaches, our approach results in faster learning of the time score and competitive or better estimation accuracies of the density ratio on challenging tasks. Furthermore, we establish theoretical guarantees on the error of the estimated density ratio.

## 1. Introduction

Estimating the ratio of two densities is a fundamental task in machine learning, with diverse applications (Sugiyama et al., 2010). For instance, by assuming that one of the densities is tractable, often a standard Gaussian, we can construct an estimator for the other density by estimating their ratio (Gutmann & Hyvärinen, 2012; Gao et al., 2019; Rhodes et al., 2020; Choi et al., 2022). It is also possible to consider a scenario where both densities are not tractable. As noted by previous works (Choi et al., 2022), density ratio estimation finds broad applications across machine learning, from mutual information estimation (Song & Ermon, 2020), generative modelling (Goodfellow et al., 2020), importance sampling (Sinha et al., 2020), likelihood-free inference (Izbicki et al., 2014) to domain adaptation (Wang et al., 2023).

---

[1]University of Helsinki, Finland [2]ENSAE, CREST, IP Paris, France. Correspondence to: Hanlin Yu <hanlin.yu@helsinki.fi>.

*Proceedings of the 42$^{nd}$ International Conference on Machine Learning*, Vancouver, Canada. PMLR 267, 2025. Copyright 2025 by the author(s).

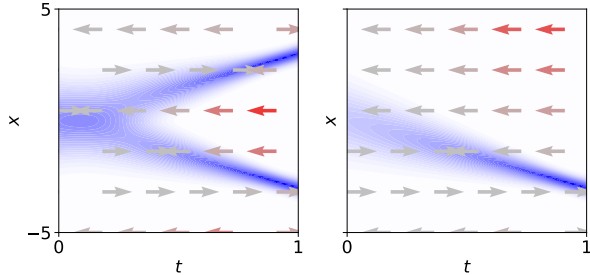

Figure 1: Densities are shown in blue. *Left*: A bi-modal probability path transitioning from a Gaussian distribution ($t = 0$) to a mixture of Diracs ($t = 1$). This path is estimated using "time scores", which are not available in closed form in general; they are depicted by arrows, with magnitudes ranging from low (gray) to high (red). *Right*: A useful decomposition of the probability path and time scores is obtained by *conditioning* on a final data point. The ensuing *conditional* density is Gaussian, and thus, the ensuing *conditional* time scores are analytically tractable. We propose to use this decomposition to estimate the "time scores".

The seminal work by Gutmann & Hyvärinen (2012) proposed a learning objective for estimating the ratio of two densities, by identifying from which density a sample is drawn. This can be done by binary classification. However, their estimator has a high variance when the densities have little overlap, which makes it impractical for problems in high dimensions (Lee et al., 2023; Chehab et al., 2023b).

To address this issue, Rhodes et al. (2020) proposed connecting the two densities with a probability path and estimating density ratios between consecutive distributions. Since two consecutive distributions are "close" to each other, the statistical efficiency may improve at the cost of increased computation, as there are multiple binary classification tasks to solve. Choi et al. (2022) examined the limiting case where the intermediate distributions become infinitesimally close. In this limit, the density ratio converges to a quantity known as the time score, which is learnt by optimizing a Time Score Matching (TSM) objective. While this limiting case leads to empirical improvements, the TSM objective is computationally inefficient to optimize, and the resulting

estimator may be inaccurate. Moreover, it is unclear what are the theoretical guarantees associated with the estimators.

In this work, we address these limitations. First, in Section 3 we introduce a novel learning objective for the time score, which we call *Conditional Time Score Matching (CTSM)*. It is based on recent advancements in generative modeling (Vincent, 2011; Pooladian et al., 2023; Tong et al., 2024a), which consider probability paths that are explicitly decomposed into mixtures of simpler paths, and where the time score is obtained in closed form. We demonstrate empirically that the CTSM objective significantly accelerates optimization in high-dimensional settings, and is several times faster compared to TSM.

Second, in Section 4 we modify our CTSM objective with a number of techniques that are popular in generative modeling (Song et al., 2021b; Choi et al., 2022; Tong et al., 2024a) to ease the learning. In particular, we derive a closed-form weighting function for the objective, as well as a vectorized version of the objective which we call *Vectorized Conditional Time Score Matching (CTSM-v)*. Together, these modifications substantially improve the estimation of the density-ratio in high dimensions, leading to stable estimators and significant speedups.

Third, in Section 5 we provide theoretical guarantees for density ratio estimation using probability paths, addressing a gap in prior works (Rhodes et al., 2020; Choi et al., 2022).

## 2. Background

Our goal is to estimate the ratio between two densities $p_0$ and $p_1$, given samples from both. We start by defining a distribution over labels $t$ and data points $\mathbf{x}$,

$$p(\mathbf{x}, t) = p(t)p(\mathbf{x} \,|\, t) \tag{1}$$

constructed such that we recover $p_0$ and $p_1$ for $t = 0$ and $t = 1$ respectively. We next show how several relevant methods can be viewed as variations on this formalism.

**Binary label**  Fundamental approaches to density-ratio estimation consider a binary label $t \in \{0, 1\}$. Among them, Noise Contrastive Estimation (NCE) is based on the observation that the density ratio is related to the binary classifier $p(t|\boldsymbol{x})$ (Gutmann & Hyvärinen, 2012, Eq. 5). NCE estimates that classifier by minimizing a binary classification loss based on logistic regression, computed using samples drawn from $p_0$ and $p_1$. In practice, using NCE is challenging when $p_0$ and $p_1$ are "far apart". In that case, both the binary classification loss becomes harder to optimize (Liu et al., 2022) and the sample-efficiency of its minimizer deteriorates (Gutmann & Hyvärinen, 2012; Lee et al., 2023; Chehab et al., 2023a;b).

**Continuous label**  More recent developments relax the label so that it is continuous $t \in [0, 1]$. Now, conditioning on $t$ defines intermediate distributions $p(\mathbf{x} \,|\, t)$, equivalently noted $p_t(\mathbf{x})$, along a probability path that connects $p_0$ to $p_1$. Then, the following identity is used (Choi et al., 2022)

$$\log \frac{p_1(\mathbf{x})}{p_0(\mathbf{x})} = \int_0^1 \partial_t \log p_t(\mathbf{x}) dt, \tag{2}$$

or its discretization in time (Rhodes et al., 2020).

**Probability path**  We next consider a popular use-case, where $p_0$ is a Gaussian and $p_1$ is the data density (Rhodes et al., 2020; Choi et al., 2022); since $p_0$ is known analytically, the ratio of the two provides directly an estimator for $p_1$. In practice, one can construct a probability path where the intermediate distributions can be sampled from but their densities cannot be evaluated. This is because the probability path is defined by interpolating samples from $p_0$ and $p_1$. There are multiple ways to define such interpolations (Rhodes et al., 2020; Albergo & Vanden-Eijnden, 2023), which we will further discuss in Section 4. A widely used approach is the Variance-Preserving (VP) probability path, which can be simulated by (Song et al., 2021b; Lipman et al., 2023; Choi et al., 2022)

$$\mathbf{x} = \sqrt{\alpha_t^2}\,\mathbf{x}_1 + \sqrt{1 - \alpha_t^2}\,\mathbf{x}_0, \tag{3}$$

where $\mathbf{x}_0 \sim \mathcal{N}(\mathbf{0}, \boldsymbol{I})$, $\mathbf{x}_1 \sim p_1$ follows the data distribution, time is drawn uniformly $t \sim \mathcal{U}[0, 1]$ and $\alpha_t \in [0, 1]$ is a positive function that increases from $0$ to $1$. By conditioning on $t$, we obtain densities $p_t(\mathbf{x}) = \frac{1}{\sqrt{1-\alpha_t^2}} p_0\left(\frac{\mathbf{x}}{\sqrt{1-\alpha_t^2}}\right) * \frac{1}{\alpha_t} p_1\left(\frac{\mathbf{x}}{\alpha_t}\right)$ that cannot be computed in closed-form, given that the density $p_1$ is unknown and that the convolution requires solving a difficult integral.

**Estimating the time score**  Importantly, the identity in Eq. 2 requires estimating the time score $\partial_t \log p_t(\mathbf{x})$, which is the Fisher score where the parameter is the label $t$. It can also be related to the binary classifier between two infinitesimally close distributions $p_t$ and $p_{t+dt}$ (Choi et al., 2022, Proposition 3). Formally, this time score can be approximated by minimizing the following *Time Score Matching (TSM)* objective

$$\mathcal{L}_{\mathrm{TSM}}(\theta) = \mathbb{E}_{p(t,\mathbf{x})}\big[\lambda(t)\big(\partial_t \log p_t(\mathbf{x}) - s_\theta(\mathbf{x}, t)\big)^2\big], \tag{4}$$

where $\lambda(t)$ is any positive weighting function. This objective requires evaluating the time score $\partial_t \log p_t(\mathbf{x})$. However, as previously explained, the formula for the time score is unavailable because the densities $p_t$, while well-defined, are not known in closed form.

To make the learning objective in Eq. 4 tractable, an insight from Hyvärinen (2005) led Choi et al. (2022); Williams et al.

(2025) to rewrite it using integration by parts. This yields

$$\mathcal{L}_{\text{TSM}}(\theta) = 2\mathbb{E}_{p_0(\mathbf{x})}[s_\theta(\mathbf{x}, 0)] - 2\mathbb{E}_{p_1(\mathbf{x})}[s_\theta(\mathbf{x}, 1)] +$$
$$\mathbb{E}_{p(t,\mathbf{x})}[2\partial_t s_\theta(\mathbf{x}, t) + 2\dot{\lambda}(t)s_\theta(\mathbf{x}, t) + \lambda(t)s_\theta(\mathbf{x}, t)^2], \quad (5)$$

which no longer requires evaluating the time score $\partial_t \log p_t(\mathbf{x})$. However, this approach has one clear computational drawback: differentiating the term $\partial_t s_\theta(\mathbf{x}, t)$ in the loss Eq. 5 involves using automatic differentiation twice — first in $t$ and then in $\theta$ — which can be time-consuming (we verify this in Section 6). This motivates us to find better ways of learning the time score.

## 3. Novel Objectives for Time Score Estimation

In this section, we propose novel methods to estimate the time score.

### 3.1. Basic Method

**Augmenting the state space** First, we rewrite Eq. 4 so that it is tractable. The idea is to further augment the state space to $(\mathbf{x}, t, \mathbf{z})$ by introducing a *conditioning variable* $\mathbf{z}$, as in related literature. Thus, we extend the model from Eq. 1 into

$$p(\mathbf{x}, t, \mathbf{z}) = p(t)p(\mathbf{z})p(\mathbf{x} \,|\, t, \mathbf{z}), \quad (6)$$

such that the intermediate distributions $p(\mathbf{x} \,|\, t, \mathbf{z})$ — now conditioned on $\mathbf{z}$ — can be sampled from *and* evaluated. We remark that this insight is shared by previous research in score matching Vincent (2011) and flow matching (Lipman et al., 2023; Pooladian et al., 2023; Tong et al., 2024a).

Consider for example Eq. 3. By choosing to condition on $\mathbf{z} = \mathbf{x}_1$, we get a closed-form $p(\mathbf{x} \,|\, t, \mathbf{z}) = \mathcal{N}(\mathbf{x}; \alpha_t \mathbf{z}, (1 - \alpha_t^2)\boldsymbol{I})$. In this example, $\mathbf{z}$ is a sample of "raw" data (for example, real observed data) while $\mathbf{x}$ is a corrupted version of data, and $t$ controls the corruption level, ranging from 0 (full corruption) to 1 (no corruption), as in Vincent (2011). In the following, we explain how to relate the descriptions of the *intractable* marginal probability path $p_t(\mathbf{x})$ to descriptions of the *tractable* conditional probability path $p_t(\mathbf{x} \,|\, \mathbf{z})$.

**Tractable objective for learning the time score** As a result of Eq. 6, we relate the time scores, obtained with and without conditioning on $\mathbf{z}$ (derivations are in Appendix D.1)

$$\partial_t \log p_t(\mathbf{x}) = \mathbb{E}_{p_t(\mathbf{z} \,|\, \mathbf{x})}[\partial_t \log p_t(\mathbf{x} \,|\, \mathbf{z})] \quad (7)$$

and exploit this identity to learn the time score, by plugging Eq. 7 into the original loss in Eq. 4. This way, we can reformulate the intractable objective in Eq. 4 into a tractable objective which we call the *Conditional Time Score Matching (CTSM)* objective

$$\mathcal{L}_{\text{CTSM}}(\theta) = \mathbb{E}_{p(\mathbf{x}, \mathbf{z}, t)}\left[\lambda(t)\left(\partial_t \log p_t(\mathbf{x} \,|\, \mathbf{z}) - s_\theta(\mathbf{x}, t)\right)^2\right]. \quad (8)$$

Note that the regression target is given by the time score of the conditional distribution, $\partial_t \log p_t(\mathbf{x} \,|\, \mathbf{z})$. The reformulation is justified by the following theorem:

**Theorem 1** (Regressing the time score) *The TSM loss Eq. 4 and CTSM loss Eq. 8 are equal, up to an additive constant.*

The proof can be found in Appendix D.2. This new objective is useful, as it requires evaluating the time score of the tractable distribution $p_t(\mathbf{x} \,|\, \mathbf{z})$ instead of the intractable distribution $p_t(\mathbf{x})$. By minimizing this objective, the model $s_\theta(\mathbf{x}, t)$ learns to output $\partial_t \log p_t(\mathbf{x})$. A similar observation was made in De Bortoli et al. (2022, Appendix L.3.), however they did not translate this observation into the CTSM objective and use it for learning. Furthermore, their setting was more restrictive, as the conditioning variable was specifically chosen to be $\mathbf{x}_1$.

### 3.2. Vectorized Variant

We propose a further objective for learning the time score, called *Vectorized Conditional Time Score Matching (CTSM-v)*. The idea is that we can easily vectorize the learning task, by forming a joint objective over the $D$ dimensions. The intuition is that the time score can be written as a sum of autoregressive terms, and that we learn each term of the sum instead of the final result only. We verify in section 6 that this approach empirically leads to better performance. Formally, define the vectorization of the conditional time score as the result of stacking its components as

$$\text{vec}(\partial_t \log p_t(\mathbf{x} \,|\, \mathbf{z})) = [\partial_t \log p_t(x^i \,|\, \mathbf{x}^{<i}, \mathbf{z})]_{i \in [\![1, D]\!]}^\top. \quad (9)$$

The time score is then obtained by summing these components. Our vectorized objective is given by

$$\mathcal{L}_{\text{CTSM-v}}(\theta) = \mathbb{E}_{p(t, \mathbf{z}, \mathbf{x})}$$
$$\left[\lambda(t)\|\text{vec}(\partial_t \log p_t(\mathbf{x} \,|\, \mathbf{z})) - \mathbf{s}_\theta^{\text{vec}}(\mathbf{x}, t)\|^2\right]. \quad (10)$$

**Theorem 2** (Regressing the vectorized time score) *The CTSM-v objective Eq. 10 is minimized when the sum of the entries of the score network equals the time score.*

This is proven in Appendix D.2. By minimizing this objective, the model $s_\theta^{\text{vec}}(\mathbf{x}, t)$ learns to output $[\mathbb{E}_{p_t(\mathbf{z} \,|\, \mathbf{x})}[\partial_t \log p_t(x^i \,|\, \mathbf{x}^{<i} \,|\, \mathbf{z})]]_{i \in [\![1, D]\!]}^\top$; this is further justified in the next Theorem 3. The original time score can be obtained from the learnt $s_\theta^{\text{vec}}(\mathbf{x}, t)$ by summing all the entries. Further, while the components of the regression target are formally given by $[\partial_t \log p_t(x^i \,|\, \mathbf{x}^{<i}, \mathbf{z})]_{i \in [\![1, D]\!]}^\top$, for commonly used probability paths like the VP path, the dependency on $\mathbf{x}^{<i}$ is dropped. We remark that Meng et al. (2020) proposed autoregressive score matching which shares similar spirit, albeit for the purpose of training scalable autoregressive models and based on Stein score.

### 3.3. General Framework

We next show that our learning objectives, i.e., both the conditional time score matching one and the vectorized variant, are actually special cases of a more general framework.

Just as we related the marginal and conditional time scores, $\partial_t \log p_t(\mathbf{x})$ and $\partial_t \log p_t(\mathbf{x} \mid \mathbf{z})$ in Eq. 7, let us now consider the same identity for general, vector or scalar valued functions $\mathbf{g}(\mathbf{x}, t)$ and $\mathbf{f}(\mathbf{x}, t, \mathbf{z})$, where $t \in [0, 1]$

$$\mathbf{g}(\mathbf{x}, t) = \mathbb{E}_{p_t(\mathbf{z} \mid \mathbf{x})}[\mathbf{f}(\mathbf{x}, t, \mathbf{z})]. \tag{11}$$

By analogy to previous paragraphs, we call the functions $\mathbf{g}$ and $\mathbf{f}$, "marginal" and "conditional". We consider the scenario where $\mathbf{g}(\mathbf{x}, t)$ is intractable, yet $\mathbf{f}(\mathbf{x}, t, \mathbf{z})$ is tractable. Similarly, we obtain a theorem that states that a regression problem over the "marginal" function $\mathbf{g}(\mathbf{x}, t)$ can be reformulated as a regression problem over the "conditional" function $\mathbf{f}(\mathbf{x}, t \mid \mathbf{z})$, thus resulting in a tractable training objective.

**Theorem 3** (Regressing a function) *Consider vector or scalar valued functions* $\mathbf{f}(\mathbf{x}, t \mid \mathbf{z})$ *and* $\mathbf{g}(\mathbf{x}, t) = \mathbb{E}_{p_t(\mathbf{z} \mid \mathbf{x})}[\mathbf{f}(\mathbf{x}, t \mid \mathbf{z})]$. *Then, the following two loss functions are equal up to an additive constant that does not depend on* $\theta$:

$$\mathcal{L}_{\mathbf{f}}(\theta) = \mathbb{E}_{p(t, \mathbf{z}, \mathbf{x})} \left[ \lambda(t) \| \mathbf{f}(\mathbf{x}, t \mid \mathbf{z}) - \mathbf{s}_\theta(\mathbf{x}, t) \|^2 \right], \tag{12}$$

$$\mathcal{L}_{\mathbf{g}}(\theta) = \mathbb{E}_{p(t, \mathbf{x})} \left[ \lambda(t) \| \mathbf{g}(\mathbf{x}, t) - \mathbf{s}_\theta(\mathbf{x}, t) \|^2 \right]. \tag{13}$$

We prove this result in Appendix D.2. Our Theorem 1 is a special case when $\mathbf{f}(\mathbf{x}, t \mid \mathbf{z}) = \partial_t \log p_t(\mathbf{x} \mid \mathbf{z})$ and $\mathbf{g}(\mathbf{x}, t) = \partial_t \log p_t(\mathbf{x})$. Similarly, our Theorem 2 is a special case when $\mathbf{f}(\mathbf{x}, t \mid \mathbf{z}) = \text{vec}(\partial_t \log p_t(\mathbf{x} \mid \mathbf{z}))$ and $\mathbf{g}(\mathbf{x}, t) = \mathbb{E}_{p_t(\mathbf{z} \mid \mathbf{x})}[\mathbf{f}(\mathbf{x}, t)]$.

Versions of Theorem 3 appear multiple times in the literature, yet they have always been stated for specific functions $\mathbf{g}$ that are Stein scores $\partial_{\mathbf{x}} \log p_t(\mathbf{x})$ (Vincent, 2011; Song et al., 2021b) or velocities that generate the probability path (Lipman et al., 2023; Pooladian et al., 2023; Tong et al., 2024a). For example, in Vincent (2011), $\mathbf{f}(\mathbf{x}, t \mid \mathbf{z}) = \partial_{\mathbf{x}} \log p_t(\mathbf{x} \mid \mathbf{z})$ and $p(t)$ is a Dirac. In Tong et al. (2024a), $\mathbf{f}(\mathbf{x}, t \mid \mathbf{z}) = v_t(\mathbf{x} \mid \mathbf{z})$ which is a velocity such that the solution to the ordinary differential equation $\dot{\mathbf{x}}_t = \mathbf{v}_t(\mathbf{x} \mid \mathbf{z})$ has marginals $p_t(\mathbf{x} \mid \mathbf{z})$. To our knowledge, it has not been stated for general functions, whose output may have any dimensionality, nor has it been applied to time scores or vectorized time scores, as we do.

## 4. Design Choices

In the previous section, we derived two novel and tractable learning objectives for the density ratio of two distributions, CTSM Eq. 8 and CTSM-v Eq. 10. In this section,

we consider two design choices for both of these learning objectives — the conditional probability path $p_t(\mathbf{x} \mid \mathbf{z})$ and the weighting function $\lambda(t)$.

**Choice of probability path** Our regression objectives require computing the time score and its vectorization of a conditional density that is analytically known. One natural choice is a Gaussian $p_t(\mathbf{x} \mid \mathbf{z}) = \mathcal{N}(\mathbf{x}; \boldsymbol{\mu}_t(\mathbf{z}), k_t \mathbf{I})$ (Lipman et al., 2023), so that the conditional time score is obtained in closed form. We specify popular choices of $\mathbf{z}, \boldsymbol{\mu}_t(\mathbf{z}), k_t$ in Appendix B.

In particular, previous works on density ratio estimation Rhodes et al. (2020); Choi et al. (2022) focused on the VP probability path Eq. 3, which is also popular in the literature of diffusion models (Sohl-Dickstein et al., 2015; Ho et al., 2020; Song et al., 2021b). By conditioning Eq. 3 on $\mathbf{z} = \mathbf{x}_1$, we obtain the conditional densities

$$p_t(\mathbf{x} \mid \mathbf{z}) = \mathcal{N}\left(\mathbf{x}; \alpha_t \mathbf{x}_1, (1 - \alpha_t^2) \mathbf{I}\right). \tag{14}$$

The conditional time score is

$$\partial_t \log p_t(\mathbf{x} \mid \mathbf{z}) = D \frac{\alpha_t \alpha_t'}{1 - \alpha_t^2} - \frac{\alpha_t \alpha_t'}{1 - \alpha_t^2} \|\epsilon\|^2 \tag{15}$$

$$+ \frac{1}{\sqrt{1 - \alpha_t^2}} \epsilon^\top \alpha_t' \mathbf{x}_1, \tag{16}$$

where $\epsilon = \frac{\mathbf{x} - \alpha_t \mathbf{x}_1}{\sqrt{1 - \alpha_t^2}}$. Finally, the vectorized conditional time score is

$$\text{vec}\left(\partial_t \log p_t(\mathbf{x} \mid \mathbf{z})\right) = \frac{\alpha_t \alpha_t'}{1 - \alpha_t^2} - \frac{\alpha_t \alpha_t'}{1 - \alpha_t^2} \epsilon^2 \tag{17}$$

$$+ \frac{1}{\sqrt{1 - \alpha_t^2}} \epsilon \, \alpha_t' \mathbf{x}_1, \tag{18}$$

where the square and the product are element-wise operations. More discussion can be found in Appendix B.1.

**Choice of weighting function** The cost function in 8 combines multiple regression tasks, indexed by $t$, into a single objective, representing a multi-task learning problem. A practical challenge is determining how to weigh the different tasks (Ruder, 2017; Rhodes et al., 2020).

Some approaches estimate a weighting function during training (Kendall et al., 2017; Nichol & Dhariwal, 2021; Choi et al., 2022; Kingma & Gao, 2023), while others use an approximation which does not depend on the parameter (Song et al., 2021b; Tong et al., 2024b). We follow the latter approach and draw inspiration from the diffusion models literature (Ho et al., 2020; Song et al., 2021b), where it is common to choose as weighting function

$$\lambda(t) \propto \frac{1}{\mathbb{E}_{p(\mathbf{x}, \mathbf{z})} \left[ \| \partial_{\mathbf{x}} \log p_t(\mathbf{x} \mid \mathbf{z}) \|^2 \right]}, \tag{19}$$

which is also the default weighting scheme from Choi et al. (2022). It was derived for estimating the Stein score $\partial_{\mathbf{x}} \log p_t(\mathbf{x} \,|\, \mathbf{z})$ (Song et al., 2021b), and we refer to this weighting scheme as *Stein score normalization*. We show in Appendix B that it simplifies to $\lambda(t) \propto k_t$.

However, as the name and the equation itself suggest, Stein score normalization is derived based on Stein score, thus not directly relating to the time score. One benefit of Stein score normalization is that its scaling essentially results in the regression targets having unit variances (Ho et al., 2020). However, the variance of the time score does not equal to the variance of the Stein score. We instead consider

$$\lambda(t) \propto \frac{1}{\mathbb{E}_{p(\mathbf{x}, \mathbf{z})}\left[\partial_t \log p_t(\mathbf{x} \,|\, \mathbf{z})^2\right]} \qquad (20)$$

for CTSM and CTSM-v. This new weighting, which we call *time score normalization*, keeps the regressands roughly equal in magnitude. We explicitly compute this novel weighting function in Appendix B: its formula depends on a quantity $c$ that is a function of the data distribution's mean and variance. A natural choice for $c$ is to compute these statistics from the data, but in our experiments, setting $c = 1$ often yields better results. In our initial experiments, we found that using the time score normalization was important to achieve stable training. We remark that it is possible to apply time score normalization Eq. 20 to CTSM-v as well: upon assuming each dimensionality having equal scales, one can calculate the variances of the objective in each individual dimension and employ the same weighting scheme.

For the specific case of the VP path Eq. 3, the time score normalization can be defined as

$$\hat{\lambda}(t) = \frac{\left(1 - \alpha_t^2\right)^2}{2\alpha_t^2 \left(\alpha_t'\right)^2 + \left(\alpha_t'\right)^2 \left(1 - \alpha_t^2\right) c}. \qquad (21)$$

**Importance sampling**  While time score normalization yields stable training in general, we empirically observe that it may not always yield the best results. Specifically, when the variance of the time score is large, for instance, when $\alpha_t \to 1$, time score normalization results in heavy down weighting. In certain cases it is beneficial to employ a weighting scheme that is approximately uniform over different values of $t$.

Inspired by diffusion models literature (Song et al., 2021a), we employ importance sampling. Specifically, samples of $t$ are drawn from another distribution $\tilde{p}(t)$,

$$\mathcal{L}(\theta) = \mathbb{E}_{p(\mathbf{x}, \mathbf{z}), \tilde{p}(t)}\left[\frac{\bar{\lambda}(t)}{\tilde{p}(t)}\left(\partial_t \log p_t(\mathbf{x} \,|\, \mathbf{z}) - s_\theta(\mathbf{x}, t)\right)^2\right], \qquad (22)$$

with the goal being that, ideally, $\frac{\bar{\lambda}(t)}{\tilde{p}(t)} = \lambda(t)$ and $\bar{\lambda}(t) \approx 1$. Further details on the employed importance sampling scheme can be found in Section C.1.

## 5. Theoretical Guarantees

In this section, we provide theoretical guarantees on the density estimated by CTSM or CTSM-v. All proofs are included in Appendix D.

In practice, we can approximate Eq. 2 as

$$\log \hat{p}_1(\mathbf{x}) = \frac{1}{K} \sum_{i=1}^{K} \hat{s}(\mathbf{x}, t_i) + \log p_0(\mathbf{x}), \qquad (23)$$

introducing two sources of error, namely the error due to discretizing the integral with $K$ steps and the error due to using the approximate time score $\hat{s}(\mathbf{x}, t)$. We quantify these errors in the following theorem. We focus on KL divergence for convenience of the derivations, and remark that the same proof can be used to bound the error between the learned and true density ratios.

**Theorem 4** (General error bound) *Denote by $p_1$ and $\hat{p}_1$ the densities obtained from Eq. 2 and Eq. 23, using the true and approximate time scores, $s(\mathbf{x}, t) := \partial_t \log p_t(\mathbf{x})$ and $\hat{s}(\mathbf{x}, t)$ respectively. Assume that the correct time score evolves smoothly with time, specifically $t \mapsto s(\mathbf{x}, t)$ is $L(\mathbf{x})$-Lipschitz. Denote as follows the time-discretized distribution $p_K(t) = \frac{1}{K} \sum_{i=1}^{K} \delta_{t_i}(t)$. The error between the two distributions $p_1$ and $\hat{p}_1$ is bounded as*

$$\mathrm{KL}(p_1, \hat{p}_1)^2 \leq \frac{1}{2K^2} \mathbb{E}_{p_1(\mathbf{x})}[L(\mathbf{x})^2]$$
$$+ 2\mathbb{E}_{p_1(\mathbf{x}), p_K(t)}\left[\left(s(\mathbf{x}, t) - \hat{s}(\mathbf{x}, t)\right)^2\right]. \qquad (24)$$

The first term quantifies a discretization error of the integral: it is null when using discretization steps $K \to \infty$, or when using paths whose time-evolution $t \to p(\mathbf{x}, t)$ is smooth, even stationary $L(\mathbf{x}) \to 0$ for any point $\mathbf{x} \in \mathbb{R}^d$ where the density is evaluated. Comparing the constants $L(\mathbf{x})$ of different probability paths is left for future work.

The second term in Eq. 24 quantifies the estimation error of the time score, collected over the times $t_i$ where it is evaluated. While such an estimation error is assumed to be constant in related works (De Bortoli et al., 2022), we specify it for both CTSM and CTSM-v in our next result.

**Proposition 5** (Error bound for CTSM and CTSM-v) *Now consider a parametric model for the time score, $s_{\boldsymbol{\theta}}(\mathbf{x}, t)$. Denote by $\theta*$ the parameter for the actual time score $\partial_t \log p_t(\mathbf{x})$, obtained by minimizing the loss from Eq. 8. Denote by $\hat{\theta}$ the parameter obtained from minimizing that same loss when the expectation is approximated using a finite sample $(\mathbf{x}_i, \mathbf{z}_i, t_i)_{i \in [\![1, N]\!]}$. Then, the expected error over all estimates $\hat{p}_1$, obtained by integrating the estimated score $s_{\hat{\theta}}(\mathbf{x}, t)$ over time, is*

$$\mathbb{E}_{\hat{p}_1}[\mathrm{KL}(p_1, \hat{p}_1)^2] \leq \frac{1}{2K^2} \mathbb{E}_{p_1(\mathbf{x})}[L(\mathbf{x})^2]$$
$$+ \frac{2}{N} e(\theta^*, \lambda, p) + o\left(\frac{1}{N}\right), \qquad (25)$$

*Note that the expectation of the* KL *is taken over all estimates* $\hat{p_1}$. *The error function* $e(\cdot)$ *is specified in Appendix D, specifically Eq. 107, with the matrices for CTSM specified in Eq. 114 and the matrices for CTSM-v specified in Eq. 121.*

Again, note that the final error decreases with the sample size $N$ and discretization steps $K$. Moreover, the estimation error of the time score depends on three design choices: the parameterization of the model $\theta \rightarrow s_\theta(\mathbf{x}, t)$, the chosen probability path $p_t(\mathbf{x} \mid \mathbf{z})$, and the weighting function $\lambda(t)$. Interestingly, there is an edge case that is *independent of the parameterization of the score* (and therefore of the choice of neural network architecture) where the error is zero. That is when the conditional and marginal scores are equal for CTSM, $\partial_t \log p_t(\mathbf{x} \mid \mathbf{z}) = \partial_t \log p_t(\mathbf{x})$, and when the vectorized version of that statement $\partial_t \log p_t(x^i \mid \mathbf{x}^{<i}, \mathbf{z}) = \mathbb{E}_{p_t(\mathbf{z} \mid \mathbf{x})} \left[ \partial_t \log p_t(x^i \mid \mathbf{x}^{<i}, \mathbf{z}) \right]$ holds true for CTSM-v. Choosing paths that approximately verify these condition could reduce the estimation error and would be interesting future work.

## 6. Experiments

To benchmark the accuracy of our CTSM objectives, we closely follow the experimental setup of Rhodes et al. (2020) and Choi et al. (2022) and also provide further experiments. Our code is available at `https://github.com/ksnxr/dre-prob-paths`.

We mainly compare with the TSM objective (Choi et al., 2022), as it was shown to outperform baseline methods like NCE (Gutmann & Hyvärinen, 2012) and TRE (Rhodes et al., 2020). Unless otherwise specified, we use the same score network, VP path, and experimental setup as in Choi et al. (2022). In these experiments, the TSM estimator is obtained using Stein score normalization as in Choi et al. (2022), while our CTSM estimators are always obtained using time score normalization; both weighting functions were defined in Section 4. In fact, we consider time score normalization an integral part of the CTSM method instead of an optional add-on, and thus do not evaluate its effect separately. Details on experiments are specified in Appendix F.

Overall, these experiments show that vectorized CTSM achieves competitive or better performance to TSM but is orders of magnitude faster, especially in higher dimensions. We note the importance of our vectorized CTSM, as in preliminary experiments, the non-vectorized CTSM is essentially not trainable on MNIST.

### 6.1. Evaluation Metrics

We follow the metrics established by prior work on density ratio estimation (Rhodes et al., 2020; Choi et al., 2022).

**Mean-Squared Error of the density ratio.** As a basic measure of estimation error, we approximate the following quantity $\mathbb{E}_{q(x)} \| \log \frac{p_1}{p_0}(x) - \widehat{\log \frac{p_1}{p_0}}(x) \|^2$ using Monte-Carlo. The distribution $q(x)$ is chosen to be the mixture $\frac{1}{2} p_0 + \frac{1}{2} p_1$ as in the implementation of Choi et al. (2022).

**Log-likelihood of the target distribution.** As a second measure of success, we approximate the following quantity $-\mathbb{E}_{p_1(\mathbf{x})}[\widehat{\log p_1}(\mathbf{x})]$ using Monte-Carlo. We report the result in bits per dimension (BPD), obtained by taking the negative log-likelihood, and then dividing by $D \log 2$ where $D$ is the dimensionality of the data.

We note that the metric of log-likelihood should be interpreted with caution. While commonly reported in related literature (Gao et al., 2019; Rhodes et al., 2020; Choi et al., 2022; Du et al., 2023), that same literature acknowledges that it is specifically designed to measure the likelihood of a normalized model. A model obtained through density-ratio estimation is only normalized in the limit of infinite samples and perfect optimization, meaning it may remain unnormalized in practice. In such cases, BPD becomes invalid because unnormalized models introduce an additive constant that distorts the BPD value. Some literature attempts to address this by re-normalizing the learned model using estimates of the log normalizing constant (Gao et al., 2019; Rhodes et al., 2020; Choi et al., 2022; Du et al., 2023). However, our experiments show these estimates can be unreliable and may even worsen the unnormalization. For example, the Annealed Importance Sampling estimator (Neal, 1998) produces highly variable log normalizing constants (e.g., ranging between $[-1100, 650]$ depending on the step size in the sampling method). Similarly, the Reverse Annealed Importance Sampling Estimator (Burda et al., 2015) can be numerically unstable for realistic distributions, such as mixtures (Du et al., 2023).

### 6.2. Model Accuracy in Synthetic Distributions with High Discrepancies

We consider synthetic data where two distributions have high discrepancies; this type of problem is considered in previous works (Choi et al., 2022) as it highlights the challenge of the density-chasm problem (Rhodes et al., 2020). For a fair comparison, we use the same model architecture, the same interpolation scheme and train for the same number of steps while tuning the learning rates for each scenario.

**Gaussians** Consider two distant Gaussians,

$$p_0(\mathbf{x}) = \mathcal{N}(\mathbf{x}; [0, \ldots, 0]^\top, \boldsymbol{I}), \quad (26)$$

$$p_1(\mathbf{x}) = \mathcal{N}(\mathbf{x}; [4, \ldots, 4]^\top, \boldsymbol{I}) \quad (27)$$

with varying dimensionality. Their density ratio is modeled by a fully-connected neural network ending with a linear

layer. Results are reported in Figure 2. We observe that our CTSM methods consistently improve upon TSM in terms of accuracy for the same number of iterations of the optimization algorithm. Moreover, a single iteration of the optimization algorithm is more than two times faster for our methods than for TSM: CTSM and CTSM-v take around 5ms per iteration, against around 15ms for TSM [1].

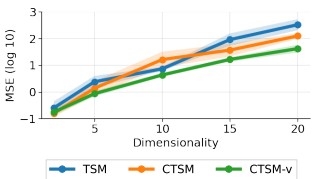

Figure 2: For estimating the density ratio between two Gaussians, CTSM-v outperforms other methods as the dimensionality increases. Full and shaded lines are respectively the means and standard deviations over 3 runs.

**Gaussian mixtures**  Consider two bi-modal Gaussian mixtures, centered at vectors of entries $\mathbf{2}$ and $-\mathbf{2}$,

$$p_0 = \frac{1}{2}\mathcal{N}(\mathbf{2} - \frac{k\sigma}{2}, \sigma^2\mathbf{I}) + \frac{1}{2}\mathcal{N}(\mathbf{2} + \frac{k\sigma}{2}, \sigma^2\mathbf{I}) \quad (28)$$

$$p_1 = \frac{1}{2}\mathcal{N}(-\mathbf{2} - \frac{k\sigma}{2}, \sigma^2\mathbf{I}) + \frac{1}{2}\mathcal{N}(-\mathbf{2} + \frac{k\sigma}{2}, \sigma^2\mathbf{I}), \quad (29)$$

with $\sigma = \sqrt{\frac{4}{4+k^2}}$. We choose the distribution in this way, such that $k$ controls the between-mode distance as a multiple of $\sigma$, while either side has unit variance in each dimension.

In this experiment specifically, the default VP path Eq. 3 cannot be used because $p_0$ is not Gaussian. We therefore use another path specified in Appendix B.2.

Results are reported in Appendix E. We observe that CTSM and CTSM-v are, again, significantly faster to run than TSM, while being able to achieve competitive performances within the same number of iterations.

### 6.3. Mutual Information Estimation for High-Dimensional Gaussians

Following Rhodes et al. (2020); Choi et al. (2022), we conduct an experiment where the goal is to estimate the mutual information between two high dimensional Gaussian distributions

$$p_0(\boldsymbol{x}) = \mathcal{N}(\mathbf{x}; \mathbf{0}, \boldsymbol{I}), \quad p_1(\boldsymbol{x}) = \mathcal{N}(\mathbf{x}; \mathbf{0}, \boldsymbol{\Sigma}), \quad (30)$$

where $\boldsymbol{\Sigma}$ is a structured matrix; specifically it is block-diagonal, where each block is $2 \times 2$ with $1$ on the diagonal

and $0.8$ on the off-diagonal, thus making the ground truth MI a function of dimensionality. Their density ratio defines the mutual information between two random variables, $\boldsymbol{x}$ restricted to even indices and $\boldsymbol{x}$ restricted to odd indices, as explained in Rhodes et al. (2020, Appendix D). Also following Rhodes et al. (2020); Choi et al. (2022), we directly parameterize a quantity related to the covariance; further details can be found in Appendix F.4.

Estimating the mutual information is a difficult task in high dimensions. Yet, as noted by Choi et al. (2022), TSM can efficiently do so. As shown in Figure 3 (right panel), all methods — TSM, CTSM and CTSM-v — can estimate the mutual information accurately after a sufficiently large number of optimization steps. However, CTSM-v is orders of magnitude faster to converge in terms of optimization step. What is more, each optimization step is consistently faster for CTSM and CTSM-v than TSM, and this effect is exacerbated in higher dimensions, as seen in Figure 3 (left panel). Overall, when running these methods with a fixed compute budget, CTSM-v outperforms both CTSM and TSM, as seen in Figure 3 (middle panel).

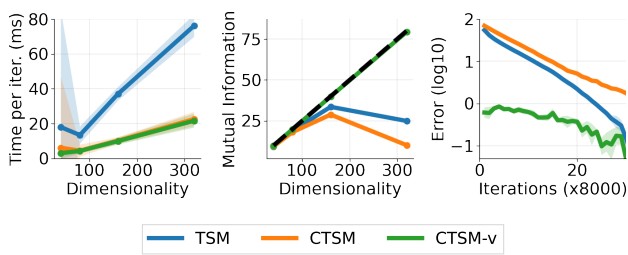

Figure 3: Mutual information estimation. *Left:* Time per iteration. *Middle:* Estimated and true (in dashed black) Mutual Information for different dimensions, where we directly report the estimates obtained after a few thousand iterations (see Appendix, Table 8). *Right:* Error between the estimated and true mutual information for dimensionality 320, during the first steps of optimization. Full and shaded lines are respectively the means and standard deviations over 3 runs.

### 6.4. Energy-based Modeling of Images

Similar to Rhodes et al. (2020) and Choi et al. (2022), we consider Energy-based Modeling (EBM) tasks on MNIST (LeCun et al., 2010). Here, we have

$$p_0(\mathbf{x}) = \mathcal{N}(\mathbf{x}; \mathbf{0}, \boldsymbol{I}), \quad p_1(\mathbf{x}) = \pi(\mathbf{x}), \quad (31)$$

where $\pi(\mathbf{x})$ is a distribution over images of digits. These images may be mapped back to an (approximately) normal distribution using a pre-trained normalizing flow (multivariate Gaussian normalizing flow).

---

[1] For this experiment, Choi et al. (2022)'s implementation of TSM had a bug (see Appendix F.1), thus the results that we report are better than the ones in their paper.

Table 1: EBM results on MNIST. Training is done either in the ambient pixel space, or in a latent space obtained using a pre-trained Gaussian normalizing flow. CTSM-v can achieve comparable results as TSM, while being much faster. For BPD, lower is better.

| Space | Methods | Approx. BPD | Time per step |
|---|---|---|---|
| Latent | TSM | 1.30 | 347 ms |
| | CTSM-v | 1.26 | 58 ms |
| Ambient | TSM | unstable | 1103 ms |
| | CTSM-v | 1.03 | 142 ms |

We note that in practice, CTSM could not be used for this task. Hence, we compare CTSM-v with TSM. To model the vectorized time score used in CTSM-v, we use the same, small U-Net architecture as in Choi et al. (2022), with one modification: to condition the network on time, we use popular Fourier feature embeddings (Tancik et al., 2020; Song et al., 2021b) instead of linear embeddings as in Choi et al. (2022). Preliminary experiments showed this led to more stable training and better final performance.

Based on preliminary experiments, we employ importance sampling to adjust the effective weighting scheme. For the implementation of the TSM loss, we directly use the original code as provided by Choi et al. (2022). We remark that the exact speed naturally depends on both the score matching algorithm and implementation details, and in our case may also depend on the way that the flow is utilized; for details we refer readers to Section F.5.

We observe that, CTSM objective can train models competitive to TSM, while being much faster. Annealed Importance Sampling, which has been used by previous works to verify the estimated log densities (Rhodes et al., 2020; Choi et al., 2022), appears to be highly unstable for time score matching algorithms, with the estimated log constants varying significantly depending on the step size of HMC algorithm.

Additionally, unlike previous related work (Rhodes et al., 2020; Choi et al., 2022), we were able to use our algorithms to successfully model MNIST images directly in the ambient pixel space. We employ as architecture a U-Net that is closer to the one used by Song et al. (2021b). We observe that these ResNet (He et al., 2016)-based architectures may result in unstable training for TSM, coinciding with the observation of Choi et al. (2022). Interestingly, the model achieves an approximate BPD value at 1.03, surpassing the best reported results in Choi et al. (2022) utilizing pre-trained flows.

**Sampling**    So far, we have used the estimated time scores to compute the target density, but they can also be used to sample from the target. To do so, we run two sampling processes, annealed MCMC and the probability flow

ODE (Song et al., 2021b): both require computing the Stein scores $\nabla \log p_t(\mathbf{x})$. Based on Eq. 2, we relate these Stein scores to the time scores

$$\nabla \log p_t(\mathbf{x}) = \nabla \int_0^t \partial_\tau \log p_\tau(\mathbf{x}) d\tau + \nabla \log p_0(\mathbf{x}).$$
(32)

Stein scores are computed by differentiating through the time scores estimated using the U-Net trained directly in the pixel space. While the time scores themselves may be well-estimated, their gradients might not be, potentially leading to inaccurate Stein scores (Liu et al., 2024). In turn, using inaccurate Stein scores in the sampling process can degrade sample quality (Chen et al., 2023). Yet, the generated samples in Figure 4 appear realistic, suggesting that, in practice, the Stein scores are well-estimated.

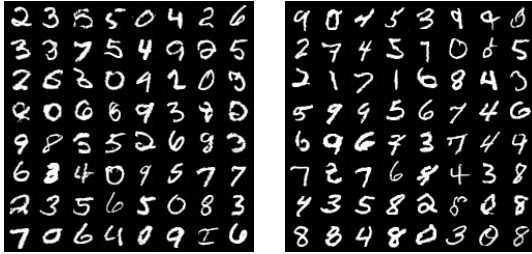

Figure 4: We report the samples obtained using ambient pixel space CTSM-v. Left: samples generated using annealed MCMC. Right: samples generated using probability flow ODE.

## 7. Discussion

**Other estimators of time score**    In this paper, we compare time score estimators based on different learning objectives. An alternative is to use a simple Monte Carlo estimator, replacing the expectation in Eq. 7 with finite samples. Similarly, Monte Carlo methods can estimate other quantities like the Stein score Scarvelis et al. (2024), though they are rarely used in practice. Recent works suggest that estimators obtained by minimizing a learning objective are preferable when the neural network architecture is well-suited to modeling the Stein score (Kadkhodaie et al., 2024; Kamb & Ganguli, 2024). A more careful exploration of these estimation methods is left for future work.

**Connections with generative modeling literature**    The learning objectives in this paper rely on probability paths that can be explicitly decomposed into mixtures of simpler probability paths. We used such simpler paths to compute the time score in closed form. Related literature has used these simpler paths to compute other quantities in closed form, such as the Stein score $\partial_\mathbf{x} \log p_t(\mathbf{x} \mid \mathbf{z})$ (Song et al.,

2021b), or the velocity (Lipman et al., 2023; Liu et al., 2023; Albergo & Vanden-Eijnden, 2023; Pooladian et al., 2023; Tong et al., 2024a) which is a vector field that transports samples from $p_0$ to $p_1$. Despite the similarities, we learn a fundamentally different quantity and our method differs from the previous ones in terms of the weighting function and an additional vectorization technique.

**Connections with multi-class classification**  Recent works have proposed to perform density ratio estimation by learning a multi-class classifier between *all* intermediate distributions, instead of multiple binary classifiers between *consecutive* intermediate distributions (Srivastava et al., 2023; Yair & Michaeli, 2023; Yadin et al., 2024). Multi-class classification seems to empirically improve the estimation of the density ratio, but compared with TSM, it has limitations in high dimensions (Srivastava et al., 2023). The limiting case where the intermediate distributions are infinitesimally close is an interesting direction for future work.

**Optimal design choices**  In this work, we introduce novel estimators of the time score that depend on many design choices. One of them is the choice of probability path. Xu et al. (2025) considered using the learned approximate optimal transport path, Wu & Xie (2025) considered using the learned approximate probability path given by annealing and Kimura & Bondell (2025) considered an information geometry formulation. Finding optimal probability paths, in the sense that the final error is minimized, is an active area of research, for example applied to estimating normalizing constants Chehab et al. (2023a),or sampling from challenging distributions (Guo et al., 2025). Another important design choice is the weighting function that has been empirically investigated in related literature (Kingma & Gao, 2023; Chen, 2023). A rigorous study of which design choice influences the final performance is left for future work.

## 8. Conclusion

We propose a new method for learning density ratios. We address a number of problems in previous work (Rhodes et al., 2020; Choi et al., 2022) that culminated in the TSM objective. First, TSM is computationally inefficient, second, the resulting estimator can be inaccurate, and third, the theoretical guarantees are not clear. Inspired by recent advances in diffusion models and flow matching, we propose the CTSM objective and directly address these three limitations. CTSM drastically reduces the running times while improving the estimation accuracy of the density ratio, especially in higher dimensions. Additionally, we develop techniques for increasing the numerical stability through, for example, novel weighting functions. Finally, we provide theoretical guarantees on the resulting estimators.

## Acknowledgements

Hanlin Yu and Arto Klami were supported by the Research Council of Finland Flagship programme: Finnish Center for Artificial Intelligence FCAI, and by the grants 345811 and 363317. Aapo Hyvärinen received funding from CIFAR. Omar Chehab and Anna Korba were supported by funding from the French ANR JCJC WOS. The authors wish to acknowledge CSC - IT Center for Science, Finland, for computational resources.

## Impact Statement

This paper presents work whose goal is to advance the field of Machine Learning. There are many potential societal consequences of our work, none of which we feel must be specifically highlighted here.

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

# Appendix

The paper and appendix are organized as follows.

## A. Useful Identities

We here organize useful identities that will be used to prove subsequent results in the form of a lemma.

**Lemma 6** (Variance of a specific random variable) *Consider two independent random variables, $\epsilon \sim \mathcal{N}(\mathbf{0}, \boldsymbol{I})$ and $\boldsymbol{x}$ with mean $\boldsymbol{\mu}$ and covariance $\boldsymbol{\Sigma}$. Then for scalars $a, b \in \mathbb{R}$,*

$$\text{Var}[a\|\epsilon\|^2 + b\,\epsilon^\top \mathbf{x}] = 2a^2 D + b^2 cD, \tag{33}$$

*where $c = (\text{Trace}\,(\boldsymbol{\Sigma}) + \|\boldsymbol{\mu}\|^2)/D$ depends on the first two moments of $\mathbf{x}$ and on the dimensionality $D$.*

*Proof of Lemma 6.* $\|\epsilon\|^2$ follows a $\chi_D^2$-distribution, which has mean $D$ and variance $2D$.

$$\mathbb{E}\left[\|\epsilon\|^4\right] = \text{Var}\left[\|\epsilon\|^2\right] + \mathbb{E}\left[\|\epsilon\|^2\right]^2 = 2D + D^2, \tag{34}$$

$$\mathbb{E}\left[\epsilon^\top \mathbf{x}\right] = \mathbb{E}\left[\sum_i \epsilon_i x_i\right] = \sum_i \mathbb{E}\left[\epsilon_i\right] \mathbb{E}\left[x_i\right] = 0, \tag{35}$$

$$\mathbb{E}\left[x_i^2\right] = \text{Var}\left[x_i\right] + \left(\mathbb{E}\left[x_i\right]\right)^2 = \Sigma_{ii} + \mu_i^2, \tag{36}$$

$$\mathbb{E}\left[\left(\epsilon^\top \mathbf{x}\right)^2\right] = \mathbb{E}\left[\sum_{i,j} \epsilon_i x_i \epsilon_j x_j\right] = \sum_{i,j} \mathbb{E}\left[\epsilon_i x_i \epsilon_j x_j\right] = \sum_i \mathbb{E}\left[\epsilon_i^2 x_i^2\right] \tag{37}$$

$$= \sum_i \mathbb{E}\left[\epsilon_i^2\right] \mathbb{E}\left[x_i^2\right] = \sum_i \left(\Sigma_{ii} + \mu_i^2\right) = \text{Tr}(\boldsymbol{\Sigma}) + \|\boldsymbol{\mu}\|^2, \tag{38}$$

$$\text{Var}\left[\epsilon^\top \mathbf{x}\right] = \mathbb{E}\left[\left(\epsilon^\top \mathbf{x}\right)^2\right] - \left(\mathbb{E}\left[\epsilon^\top \mathbf{x}\right]\right)^2 = \mathbb{E}\left[\left(\epsilon^\top \mathbf{x}\right)^2\right] = \text{Tr}(\boldsymbol{\Sigma}) + \|\boldsymbol{\mu}\|^2, \tag{39}$$

$$\mathbb{E}\left[\|\epsilon\|^2 \epsilon^\top \mathbf{x}\right] = \mathbb{E}\left[\left(\sum_i \epsilon_i^2\right) \sum_j \epsilon_j x_j\right] = \mathbb{E}\left[\sum_j \epsilon_j^3 x_j\right] + \mathbb{E}\left[\left(\sum_{i \neq j} \epsilon_i^2\right) \sum_j \epsilon_j x_j\right] \tag{40}$$

$$= \left(\sum_j \mathbb{E}\left[\epsilon_j^3\right]\right) \mathbb{E}\left[x_j\right] + \mathbb{E}\left[\sum_{i \neq j} \epsilon_i^2\right] \sum_j \mathbb{E}\left[\epsilon_j\right] \mathbb{E}\left[x_j\right] = 0, \tag{41}$$

$$\text{Var}\left[a\|\epsilon\|^2 + b\,\epsilon^\top \mathbf{x}\right] = \mathbb{E}\left[\left(a\|\epsilon\|^2 + b\,\epsilon^\top \mathbf{x}\right)^2\right] - \left(\mathbb{E}\left[a\|\epsilon\|^2 + b\,\epsilon^\top \mathbf{x}\right]\right)^2 \tag{42}$$

$$= \mathbb{E}\left[a^2\|\epsilon\|^4 + 2ab\|\epsilon\|^2 \epsilon^\top \mathbf{x} + b^2\left(\epsilon^\top \mathbf{x}\right)^2\right] - \left(\mathbb{E}\left[a\|\epsilon\|^2 + b\,\epsilon^\top \mathbf{x}\right]\right)^2 \tag{43}$$

$$= a^2\left(2D + D^2\right) + 0 + b^2\left(\text{Tr}(\boldsymbol{\Sigma}) + \|\boldsymbol{\mu}\|^2\right) - (aD)^2 = 2a^2 D + b^2\left(\text{Tr}(\boldsymbol{\Sigma}) + \|\boldsymbol{\mu}\|^2\right) \tag{44}$$

$$= 2a^2 D + b^2 cD. \tag{45}$$

$\square$

## B. Probability Paths

**Definition**   In this paper, we consider probability paths $p_t(\mathbf{x})$ that are explicitly decomposed as a mixture of simpler probability paths $p_t(\mathbf{x} \mid \mathbf{z})$, where $\mathbf{z}$ indexes the mixture. Formally, this is written as

$$p_t(\mathbf{x}) = \mathbb{E}_{p(\mathbf{z})}[p_t(\mathbf{x} \mid \mathbf{z})] = \mathbb{E}_{p(\mathbf{z})}\left[\mathcal{N}(\mathbf{x}; \boldsymbol{\mu}_t(\mathbf{z}), k_t \, \mathbf{I})\right]. \tag{46}$$

The conditional paths are chosen to be Gaussian $p_t(\mathbf{x} \mid \mathbf{z}) = \mathcal{N}(\mathbf{x}; \boldsymbol{\mu}_t(\mathbf{z}), k_t \, \mathbf{I})$. We will specify popular choices of $\mathbf{z}$, $\boldsymbol{\mu}_t(\mathbf{z})$, and $k_t$ in Sections B.1 and B.2.

**Time score**   We have

$$\log p_t(\mathbf{x} \mid \mathbf{z}) = -\frac{1}{2}\log\det(k_t \, \mathbf{I}) - \frac{1}{2}(\mathbf{x} - \boldsymbol{\mu}_t)^\top k_t^{-1}(\mathbf{x} - \boldsymbol{\mu}_t) + \text{const} \tag{47}$$

$$= -\frac{1}{2}\log(k_t^D) - \frac{1}{2k_t}\|\mathbf{x} - \boldsymbol{\mu}_t\|^2 + \text{const} \tag{48}$$

$$= -\frac{1}{2}D\log(k_t) - \frac{1}{2k_t}\|\mathbf{x} - \boldsymbol{\mu}_t\|^2 + \text{const}, \tag{49}$$

$$\partial_t \log p_t(\mathbf{x} \mid \mathbf{z}) = -\frac{1}{2}D\frac{\partial_t k_t}{k_t} - \frac{1}{2}\left(-\frac{\partial_t k_t}{k_t^2}\|\mathbf{x} - \boldsymbol{\mu}_t\|^2 + k_t^{-1}2(\mathbf{x} - \boldsymbol{\mu}_t)^\top(-\partial_t \, \boldsymbol{\mu}_t)\right) \tag{50}$$

$$= -\frac{1}{2}D\frac{\partial_t k_t}{k_t} + \frac{\partial_t k_t}{2k_t^2}\|\mathbf{x} - \boldsymbol{\mu}_t\|^2 + \frac{1}{k_t}(\mathbf{x} - \boldsymbol{\mu}_t)^\top \partial_t \, \boldsymbol{\mu}_t. \tag{51}$$

As such, the time score is

$$\partial_t \log p_t(\mathbf{x} \mid \mathbf{z}) = \frac{-D\dot{k}_t}{2k_t} + \frac{1}{\sqrt{k_t}}\dot{\boldsymbol{\mu}}_t^\top \, \epsilon_t(\mathbf{x}, \mathbf{z}) + \frac{\dot{k}_t}{2k_t}\|\epsilon_t(\mathbf{x}, \mathbf{z})\|^2, \qquad \epsilon_t(\mathbf{x}, \mathbf{z}) = \frac{1}{\sqrt{k_t}}(\mathbf{x} - \boldsymbol{\mu}_t(\mathbf{z})). \tag{52}$$

In fact, we can formally write the time score without the conditioning variable,

$$\partial_t \log p_t(\mathbf{x}) = \mathbb{E}_{p_t(\mathbf{z} \mid \mathbf{x})}[\partial_t \log p_t(\mathbf{x} \mid \mathbf{z})], \quad p_t(\mathbf{z} \mid \mathbf{x}) \propto p(\mathbf{z})\exp\left(-\frac{1}{2k_t}\|\mathbf{x} - \boldsymbol{\mu}_t(\mathbf{z})\|^2\right). \tag{53}$$

**Vectorized time score**   The vectorized version is simply given by

$$\text{vec}(\partial_t \log p_t(\mathbf{x} \mid \mathbf{z})) = \frac{-\dot{k}_t}{2k_t} + \frac{1}{\sqrt{k_t}}\dot{\boldsymbol{\mu}}_t \, \epsilon_t(\mathbf{x}, \mathbf{z}) + \frac{\dot{k}_t}{2k_t}\epsilon_t(\mathbf{x}, \mathbf{z})^2, \tag{54}$$

which can be obtained by decomposing the time score as the sum of $D$ terms.

**Stein score**   The Stein score is (Kingma & Gao, 2023)

$$\partial_{\mathbf{x}} \log p_t(\mathbf{x} \mid \mathbf{z}) = -\frac{1}{\sqrt{k_t}}\epsilon_t(\mathbf{x}, \mathbf{z}), \qquad \epsilon_t(\mathbf{x}, \mathbf{z}) = \frac{1}{\sqrt{k_t}}(\mathbf{x} - \boldsymbol{\mu}_t(\mathbf{z})). \tag{55}$$

**Stein score normalization**   Observe that for a fixed $t$, $\epsilon$ is, by definition, sampled from a standard normal distribution. As such, the Stein score in Equation Eq. 55 has variance $\frac{1}{k_t}$. The Stein score normalization in Eq. 19 is therefore given by

$$\lambda(t) \propto k_t. \tag{56}$$

### B.1. Variance-Preserving Probability Path

**Simulating the path**   This path is simulated by interpolating the random variables $(\mathbf{x}_0, \mathbf{x}_1) \sim p_0 \otimes p_1$,

$$\mathbf{x} = \alpha_t \, \mathbf{x}_1 + \sqrt{1 - \alpha_t^2} \, \mathbf{x}_0. \tag{57}$$

**Definition** Conditioning on $t$ and $\mathbf{z} = \mathbf{x}_1$, and choosing a Gaussian reference distribution $p_0(\mathbf{x}) = \mathcal{N}(\mathbf{x}; 0, I)$, yields

$$\boldsymbol{\mu}_t(\mathbf{z}) = \alpha_t \mathbf{x}_1, \quad k_t = 1 - \alpha_t^2. \tag{58}$$

These choices define a popular probability path, sometimes called "variance-preserving" as the variance of $p_t(\boldsymbol{x})$ is constant for all $t \in [0, 1]$ (Sohl-Dickstein et al., 2015; Ho et al., 2020; Song et al., 2021b; Lipman et al., 2023). This path is in fact the default choice in the work most related to ours (Choi et al., 2022). In the above, $\alpha_t$ is positive and increasing, such that $\alpha_0 = 0$ and $\alpha_1 = 1$. It is sometimes referred to as the noise schedule (Chen, 2023). Popular choices include exponential $\alpha_t = \min(1, e^{-2(T-t)})$ (Song et al., 2021b) for some fixed $T \geq 0$, or linear $\alpha_t = \min(1, t)$ functions (Albergo & Vanden-Eijnden, 2023; Gao et al., 2024).

We remark that in diffusion models literature (Song et al., 2021b), $p_0$ denotes data and $p_1$ denotes noise. We follow the flow matching convention, and use $p_0$ to denote noise and $p_1$ to denote data.

**Stein score** The resulting Stein score from Eq. 55 is

$$\partial_{\mathbf{x}} \log p(\mathbf{x} \mid \mathbf{z}) = -\frac{1}{\sqrt{1 - \alpha_t^2}} \, \epsilon_t(\mathbf{x}, \mathbf{z}). \tag{59}$$

**Stein score normalization** We have

$$\lambda(t) \propto 1 - \alpha_t^2. \tag{60}$$

**Time score** The resulting time score from Eq. 52 is

$$\partial_t \log p_t(\mathbf{x} \mid \mathbf{z}) = D \frac{\alpha_t \alpha_t'}{1 - \alpha_t^2} - \frac{\alpha_t \alpha_t'}{(1 - \alpha_t^2)^2} \|\mathbf{x} - \alpha_t \mathbf{x}_1\|^2 + \frac{1}{1 - \alpha_t^2} (\mathbf{x} - \alpha_t \mathbf{x}_1)^\top \alpha_t' \mathbf{x}_1 \tag{61}$$

$$= D \frac{\alpha_t \alpha_t'}{1 - \alpha_t^2} - \frac{\alpha_t \alpha_t'}{1 - \alpha_t^2} \|\epsilon\|^2 + \frac{1}{\sqrt{1 - \alpha_t^2}} \epsilon^\top \alpha_t' \mathbf{x}_1. \tag{62}$$

**Vectorized time score** The vectorized version of the time score can be obtained as

$$\partial_t \log p_t(\mathbf{x} \mid \mathbf{z}) = D \frac{\alpha_t \alpha_t'}{1 - \alpha_t^2} - \frac{\alpha_t \alpha_t'}{(1 - \alpha_t^2)^2} (\mathbf{x} - \alpha_t \mathbf{x}_1)^2 + \frac{1}{1 - \alpha_t^2} (\mathbf{x} - \alpha_t \mathbf{x}_1) \alpha_t' \mathbf{x}_1 \tag{63}$$

$$= D \frac{\alpha_t \alpha_t'}{1 - \alpha_t^2} - \frac{\alpha_t \alpha_t'}{1 - \alpha_t^2} \epsilon^2 + \frac{1}{\sqrt{1 - \alpha_t^2}} \epsilon \, \alpha_t' \mathbf{x}_1. \tag{64}$$

**Time score normalization** We have

$$\mathrm{Var}_{p_t(\mathbf{z}, \mathbf{x})}[\partial_t \log p_t(\mathbf{x} \mid \mathbf{z})] = \frac{2\alpha_t^2 (\alpha_t')^2 + (\alpha_t')^2 (1 - \alpha_t^2) c}{(1 - \alpha_t^2)^2}. \tag{65}$$

where $c = (\mathrm{Trace}(\boldsymbol{\Sigma}) + \|\boldsymbol{\mu}\|^2)/D$ depends on the mean $\boldsymbol{\mu}$, covariance $\boldsymbol{\Sigma}$ and dimensionality $D$ of $\mathbf{x}$.

To compute the variance of the time score, observe that the first term is deterministic and therefore does not participate in the computation of the variance. To obtain the variance of the two remaining terms, we apply Lemma 6 with $a = -\frac{\alpha(t)\alpha'(t)}{1 - \alpha(t)^2}$ and $b = \frac{1}{\sqrt{1 - \alpha(t)^2}}$.

Interestingly, the variance can explode $\mathrm{Var}[\partial_t \log p_t(\mathbf{x} \mid \mathbf{z})] \to \infty$ near the target distribution $\alpha(t) \to 1$.

### B.2. Schrödinger Bridge Probability Path

**Simulating the path** This path is simulated by interpolating the random variables $(\mathbf{x}_0, \mathbf{x}_1) \sim \pi(\mathbf{x}_0, \mathbf{x}_1)$, generated from a coupling $\pi$ of the marginals $p_0$ and $p_1$, and adding Gaussian noise $\epsilon \sim \mathcal{N}(\mathbf{0}, \boldsymbol{I})$ between the endpoints,

$$\mathbf{x} = t \mathbf{x}_1 + (1 - t) \mathbf{x}_0 + \sigma \sqrt{t(1 - t)} \, \epsilon. \tag{66}$$

**Definition** Conditioning on $t$ and $\mathbf{z} = (\mathbf{x}_1, \mathbf{x}_0)$, yields

$$\mu_t(z) = (1 - t)\,\mathbf{x}_0 + t\,\mathbf{x}_1, \quad k_t = t(1 - t)\sigma^2. \tag{67}$$

These choices define another path of distributions in the literature. Typically, the coupling from which $\mathbf{z}$ is drawn is either the product distribution $p_0 \otimes p_1$ or a coupling $\pi$ that satisfies optimal transport. In the latter case, the ensuing path is known as a Schrödinger bridge (Föllmer, 1988; Tong et al., 2024b). In practice, the optimal transport coupling can be approximated using limited samples from both $p_0$ and $p_1$ (Pooladian et al., 2023; Tong et al., 2024a). For simplicity, we use the product distribution. Note that for this path, $p_0$ need not be a Gaussian.

When using independent couplings with $p_0$ and $p_1$ having equal variance $var$, arguably the most natural choice of $\sigma$ is to set $\sigma = \sqrt{2var}$. In this case, the variance is preserved along the path. In order to see that, observe that under this setting the variance of $\mathbf{x}_t$ is given by

$$t^2 var + (1 - t)^2 var + 2t(1 - t)var = var.$$

However, empirically one may achieve better results with other choices of $\sigma$.

**Stein score** The resulting Stein score from Eq. 55 is

$$\partial_{\mathbf{x}} \log p(\mathbf{x} \mid \mathbf{z}) = -\frac{1}{\sigma\sqrt{t(1 - t)}}\,\epsilon_t(\mathbf{x}, \mathbf{z}). \tag{68}$$

**Stein score normalization** The Stein score normalization is given by

$$\lambda(t) \propto t(1 - t)\sigma^2. \tag{69}$$

**Time score** The resulting time score from Eq. 52 is

$$\partial_t \log p_t(\mathbf{x} \mid \mathbf{z}) = -\frac{1}{2}D\frac{1 - 2t}{t(1 - t)} + \frac{1 - 2t}{2\left(t(1 - t)\right)^2 \sigma^2}\left\|\mathbf{x} - (1 - t)\,\mathbf{x}_0 - t\,\mathbf{x}_1\right\|^2 \tag{70}$$

$$+ \frac{1}{t(1 - t)\sigma^2}\left(\mathbf{x} - (1 - t)\,\mathbf{x}_0 - t\,\mathbf{x}_1\right)^\top (\mathbf{x}_1 - \mathbf{x}_0) \tag{71}$$

$$= -\frac{1}{2}D\frac{1 - 2t}{t(1 - t)} + \frac{1 - 2t}{2t(1 - t)}\|\epsilon\|^2 + \frac{1}{\sqrt{t(1 - t)}\sigma}\epsilon^\top (\mathbf{x}_1 - \mathbf{x}_0). \tag{72}$$

**Vectorized time score** The vectorized time score is given by

$$\mathrm{vec}(\partial_t \log p_t(\mathbf{x} \mid \mathbf{z})) = -\frac{1}{2}D\frac{1 - 2t}{t(1 - t)} + \frac{1 - 2t}{2\left(t(1 - t)\right)^2 \sigma^2}\left(\mathbf{x} - (1 - t)\,\mathbf{x}_0 - t\,\mathbf{x}_1\right)^2 \tag{73}$$

$$+ \frac{1}{t(1 - t)\sigma^2}\left(\mathbf{x} - (1 - t)\,\mathbf{x}_0 - t\,\mathbf{x}_1\right)(\mathbf{x}_1 - \mathbf{x}_0) \tag{74}$$

$$= -\frac{1}{2}D\frac{1 - 2t}{t(1 - t)} + \frac{1 - 2t}{2t(1 - t)}\epsilon^2 + \frac{1}{\sqrt{t(1 - t)}\sigma}\epsilon\,(\mathbf{x}_1 - \mathbf{x}_0). \tag{75}$$

**Time score normalization** To compute the variance, treat $\frac{\mathbf{x}_1 - \mathbf{x}_0}{\sigma}$ as a random variable with mean $\boldsymbol{\mu}$ and covariance $\boldsymbol{\Sigma}$, we observe that, similar to VP path, it can be written as $a = \frac{1 - 2t}{2t(1 - t)}$ and $b = \frac{1}{\sqrt{t(1 - t)}}$.

We have

$$\mathrm{Var}_{p_t(\mathbf{z},\mathbf{x})}[\partial_t \log p_t(\mathbf{x} \mid \mathbf{z})] = \frac{1 - 4t + 4t^2 + 2ct - 2ct^2}{2t^2(1 - t)^2}D. \tag{76}$$

Note that as $t$ approaches 0 or 1, the variance may be infinite.

# C. Weighting Scheme

## C.1. Details on Importance Sampling

We consider the simple VP path, given by $\mathbf{x} = t\,\mathbf{x}_1 + \sqrt{1-t^2}\,\mathbf{x}_0$, where $\mathbf{x}_0$ is standard Gaussian, and $\mathbf{x}_1$ is a distribution with $c = 1$. One divided by the time score normalization is given by $\frac{1+t^2}{(1-t^2)^2}$. Treating this as an unnormalized probability density defined between $0$ and $t_1$, one can derive that the normalization constant is given by $Z = \frac{t_1}{1-t_1^2}$, and the CDF is given by $y(t) = \frac{1}{Z}\frac{t}{1-t^2}$. We can calculate the inverse CDF as

$$\frac{-1+\sqrt{1+4y^2Z^2}}{2yZ} = \frac{2yZ}{\sqrt{1+4y^2Z^2}+1}. \tag{77}$$

As such, we can draw samples between $0$ and $t_1$ using the inverse CDF transform. Re-normalize $t_1$ to lie between $0$ and $1-\epsilon$ yields the final samples.

In practice, we choose $t_1 = 0.9$ and employ this heuristic scheme for EBM experiments though we are using a different variant of the VP path.

# D. Theoretical Results

## D.1. Proof of Eq. 7

*Proof of Eq. 7.* The derivations are similar to denoising score matching (Vincent, 2011; Bortoli et al., 2024).

We wish to relate the time score $\partial_t \log p_t(\mathbf{x})$ and the conditional time score $\partial_t \log p_t(\mathbf{x}\,|\,\mathbf{z})$.

We have

$$p_t(\mathbf{x}) = \int p_t(\mathbf{x}\,|\,\mathbf{z})p(\mathbf{z})\mathrm{d}\,\mathbf{z}, \tag{78}$$

$$\partial_t p_t(\mathbf{x}) = \int \partial_t p_t(\mathbf{x}\,|\,\mathbf{z})p(\mathbf{z})\mathrm{d}\,\mathbf{z} = \int \partial_t \log p_t(\mathbf{x}\,|\,\mathbf{z})p_t(\mathbf{x}\,|\,\mathbf{z})p(\mathbf{z})\mathrm{d}\,\mathbf{z}, \tag{79}$$

therefore

$$\partial_t \log p_t(\mathbf{x}) = \frac{\partial_t p_t(\mathbf{x})}{p_t(\mathbf{x})} = \int \partial_t \log p_t(\mathbf{x}\,|\,\mathbf{z})\frac{p_t(\mathbf{x}\,|\,\mathbf{z})p(\mathbf{z})}{p_t(\mathbf{x})}\mathrm{d}\,\mathbf{z} = \int \partial_t \log p_t(\mathbf{x}\,|\,\mathbf{z})p_t(\mathbf{z}\,|\,\mathbf{x})\mathrm{d}\,\mathbf{z}. \tag{80}$$

$\square$

## D.2. Proofs of Theorems 1, 2 and 3

We note that Theorems 1 and 2 are special cases of Theorem 3.

For Theorem 1, $\mathbf{f}(\mathbf{x}, t\,|\,\mathbf{z}) = \partial_t \log p_t(\mathbf{x}\,|\,\mathbf{z})$, in which case $\mathbf{g}(\mathbf{x}, t) = \mathbb{E}_{p_t(\mathbf{z}\,|\,\mathbf{x})}\left[\partial_t \log p_t(\mathbf{x}\,|\,\mathbf{z})\right] = \partial_t \log p_t(\mathbf{x})$, i.e. the time score itself.

For Theorem 2, $\mathbf{f}(\mathbf{x}, t\,|\,\mathbf{z}) = \mathrm{vec}(\partial_t \log p_t(\mathbf{x}\,|\,\mathbf{z}))$, in which case $\mathbf{g}(\mathbf{x}, t) = \mathbb{E}_{p_t(\mathbf{z}\,|\,\mathbf{x})}\left[\mathrm{vec}(\partial_t \log p_t(\mathbf{x}\,|\,\mathbf{z}))\right]$. It is clear that

$$\sum_i \mathbb{E}_{p_t(\mathbf{z}\,|\,\mathbf{x})}\left[\partial_t \log p_t(x^i\,|\,\mathbf{x}^{<i}, \mathbf{z})\right] = \mathbb{E}_{p_t(\mathbf{z}\,|\,\mathbf{x})}\left[\sum_i \partial_t \log p_t(x^i\,|\,\mathbf{x}^{<i}, \mathbf{z})\right] = \mathbb{E}_{p_t(\mathbf{z}\,|\,\mathbf{x})}\left[\partial_t \log p_t(\mathbf{x}\,|\,\mathbf{z})\right] = \partial_t \log p_t(\mathbf{x}), \tag{81}$$

i.e. the sum of $\mathbf{g}(\mathbf{x}, t)$ gives the time score.

We prove Theorem 3 in what follows.

*Proof of Theorem 3.* The derivations are similar to Lipman et al. (2023); Tong et al. (2024a). First, we compute the gradients

of both cost functions, $J_{\mathbf{g}}$ and $J_{\mathbf{f}}$.

$$\nabla_\theta J_{\mathbf{g}}(\theta) = \nabla_\theta \mathbb{E}_{p(t),p_t(\mathbf{x})} \left[ \lambda(t) \|\mathbf{g}(\mathbf{x},t) - \mathbf{s}_\theta(\mathbf{x},t)\|^2 \right] \tag{82}$$

$$= \nabla_\theta \mathbb{E}_{p(t),p_t(\mathbf{x})} \left[ \lambda(t) \left( \|\mathbf{g}(\mathbf{x},t)\|^2 - 2 \langle \mathbf{g}(\mathbf{x},t), \mathbf{s}_\theta(\mathbf{x},t) \rangle + \|\mathbf{s}_\theta(\mathbf{x},t)\|^2 \right) \right] \tag{83}$$

$$= \nabla_\theta \mathbb{E}_{p(t),p_t(\mathbf{x})} \left[ \lambda(t) \left( -2 \langle \mathbf{g}(\mathbf{x},t), \mathbf{s}_\theta(\mathbf{x},t) \rangle + \|\mathbf{s}_\theta(\mathbf{x},t)\|^2 \right) \right]. \tag{84}$$

$$\nabla_\theta J_{\mathbf{f}}(\theta) = \nabla_\theta \mathbb{E}_{p(t),p(\mathbf{z}),p_t(\mathbf{x}\,|\,\mathbf{z})} \left[ \lambda(t) \|\mathbf{f}(\mathbf{x},t\,|\,\mathbf{z}) - \mathbf{s}_\theta(\mathbf{x},t)\|^2 \right] \tag{85}$$

$$= \nabla_\theta \mathbb{E}_{p(t),p(\mathbf{z}),p_t(\mathbf{x}\,|\,\mathbf{z})} \left[ \lambda(t) \left( \|\mathbf{f}(\mathbf{x},t\,|\,\mathbf{z})\|^2 - 2 \langle \mathbf{f}(\mathbf{x},t\,|\,\mathbf{z}), \mathbf{s}_\theta(\mathbf{x},t) \rangle + \|\mathbf{s}_\theta(\mathbf{x},t)\|^2 \right) \right] \tag{86}$$

$$= \nabla_\theta \mathbb{E}_{p(t),p(\mathbf{z}),p_t(\mathbf{x}\,|\,\mathbf{z})} \left[ \lambda(t) \left( -2 \langle \mathbf{f}(\mathbf{x},t\,|\,\mathbf{z}), \mathbf{s}_\theta(\mathbf{x},t) \rangle + \|\mathbf{s}_\theta(\mathbf{x},t)\|^2 \right) \right]. \tag{87}$$

We then proceed to show that the two terms coincide:

$$\mathbb{E}_{p_t(\mathbf{x})} \|\mathbf{s}_\theta(\mathbf{x},t)\|^2 = \mathbb{E}_{p(\mathbf{z})p_t(\mathbf{x}\,|\,\mathbf{z})} \|\mathbf{s}_\theta(\mathbf{x},t)\|^2, \tag{88}$$

$$\mathbf{g}(\mathbf{x},t) = \int \frac{p_t(\mathbf{x}\,|\,\mathbf{z})p(\mathbf{z})}{p_t(\mathbf{x})} \mathbf{f}(\mathbf{x},t\,|\,\mathbf{z}) \mathrm{d}\,\mathbf{z}, \tag{89}$$

$$\mathbb{E}_{p_t(\mathbf{x})} \langle \mathbf{g}(\mathbf{x},t), \mathbf{s}_\theta(\mathbf{x},t) \rangle = \mathbb{E}_{p_t(\mathbf{x})} \left\langle \int \frac{p_t(\mathbf{x}\,|\,\mathbf{z})p(\mathbf{z})}{p_t(\mathbf{x})} \mathbf{f}(\mathbf{x},t\,|\,\mathbf{z}) \mathrm{d}\,\mathbf{z}, \mathbf{s}_\theta(\mathbf{x},t) \right\rangle \tag{90}$$

$$= \int \left\langle \int \frac{p_t(\mathbf{x}\,|\,\mathbf{z})p(\mathbf{z})}{p_t(\mathbf{x})} \mathbf{f}(\mathbf{x},t\,|\,\mathbf{z}) \mathrm{d}\,\mathbf{z}, \mathbf{s}_\theta(\mathbf{x},t) \right\rangle p_t(\mathbf{x}) \mathrm{d}\,\mathbf{x} \tag{91}$$

$$= \int \left\langle \int p_t(\mathbf{x}\,|\,\mathbf{z})p(\mathbf{z}) \mathbf{f}(\mathbf{x},t\,|\,\mathbf{z}) \mathrm{d}\,\mathbf{z}, \mathbf{s}_\theta(\mathbf{x},t) \right\rangle \mathrm{d}\,\mathbf{x} \tag{92}$$

$$= \int \int \langle \mathbf{f}(\mathbf{x},t\,|\,\mathbf{z}), \mathbf{s}_\theta(\mathbf{x},t) \rangle p_t(\mathbf{x}\,|\,\mathbf{z})p(\mathbf{z}) \mathrm{d}\,\mathbf{z}\, \mathrm{d}\,\mathbf{x}. \tag{93}$$

$$\square$$

### D.3. Proof of Theorem 4

*Proof of Theorem 4.* We have

$$\mathrm{KL}(p_1,\hat{p}_1)^2 = \left( \mathbb{E}_{p_1(\mathbf{x})} [\log p_1(\mathbf{x}) - \log \hat{p}_1(\mathbf{x})] \right)^2 \tag{94}$$

$$\leq \mathbb{E}_{p_1(\mathbf{x})} \left[ (\log p_1(\mathbf{x}) - \log \hat{p}_1(\mathbf{x}))^2 \right] \tag{95}$$

$$= \mathbb{E}_{p_1(\mathbf{x})} \left[ \left( \int_0^1 s(\mathbf{x},t)dt - \frac{1}{K} \sum_{i=1}^K \hat{s}(\mathbf{x},t_i) \right)^2 \right] \tag{96}$$

$$= \mathbb{E}_{p_1(\mathbf{x})} \left[ \left( \int_0^1 s(\mathbf{x},t)dt - \frac{1}{K} \sum_{i=1}^K s(\mathbf{x},t_i) + \frac{1}{K} \sum_{i=1}^K s(\mathbf{x},t_i) - \frac{1}{K} \sum_{i=1}^K \hat{s}(\mathbf{x},t_i) \right)^2 \right] \tag{97}$$

$$\leq \mathbb{E}_{p_1(\mathbf{x})} \left[ 2 \left( \int_0^1 s(\mathbf{x},t)dt - \frac{1}{K} \sum_{i=1}^K s(\mathbf{x},t_i) \right)^2 + 2 \left( \frac{1}{K} \sum_{i=1}^K s(\mathbf{x},t_i) - \frac{1}{K} \sum_{i=1}^K \hat{s}(\mathbf{x},t_i) \right)^2 \right] \tag{98}$$

$$\leq \mathbb{E}_{p_1(\mathbf{x})} \left[ 2 \left( \frac{L(\mathbf{x})}{2K} \right)^2 + 2 \frac{1}{K} \sum_{i=1}^K (s(\mathbf{x},t_i) - \hat{s}(\mathbf{x},t_i))^2 \right] \tag{99}$$

$$= \frac{1}{2K^2} \mathbb{E}_{p_1(\mathbf{x})}[L(\mathbf{x})^2] + 2 \mathbb{E}_{p_1(\mathbf{x}),p_K(t)} \left[ (s(\mathbf{x},t) - \hat{s}(\mathbf{x},t))^2 \right], \tag{100}$$

where we used Jensen's inequality and bound the discretization error of a Riemannian integral using the left rectangular sum.

$$\square$$

### D.4. Proof of Proposition 5

*Proof of Proposition 5.* Denote by $s(\mathbf{x}, \mathbf{z}, t) = \partial_t \log p_t(\mathbf{x} \,|\, \mathbf{z})$ the conditional score and by $l_\theta(\mathbf{x}, \mathbf{z}, t) = \lambda(t)\left(s(\mathbf{x}, \mathbf{z}, t) - s_\theta(\mathbf{x}, t)\right)^2$. The population and empirical losses defined from Eq. 8 are respectively

$$\mathcal{L}_{\text{CTSM}}(\theta) = \mathbb{E}_{p(t, \mathbf{x}, \mathbf{z})}[l_\theta(\mathbf{x}, \mathbf{z}, t)], \qquad \hat{\mathcal{L}}_{\text{CTSM}}(\theta) = \frac{1}{N} \sum_{i=1}^{N} l_\theta(\mathbf{x}_i, \mathbf{z}_i, t_i), \tag{101}$$

where the empirical loss uses *i.i.d.* samples $(\mathbf{x}_i, \mathbf{z}_i, t_i)_{i \in [\![1, N]\!]}$. In the following, we suppose that the model is well-specified, which means that there exists a $\theta^*$ that parameterizes the true score.

**Error formulas**    First, we compute the error in the parameters. Using Bach (2024, Section 4.7) and van der Vaart (2000, Theorem 5.23),

$$\sqrt{N}(\hat{\theta} - \theta^*) \sim \mathcal{N}(0, H(\theta^*)^{-1} G(\theta^*) H(\theta^*)^{-1}), \tag{102}$$

where $G(\theta^*)$ and $H(\theta^*)$ are matrices that will be later specified.

Then, we obtain the error in the scores, using the delta method

$$\sqrt{N}(s_{\hat{\theta}}(\mathbf{x}, t) - s_{\theta^*}(\mathbf{x}, t)) \sim \mathcal{N}(0, \nabla_\theta s_\theta(\mathbf{x}, t)|_{\theta^*}^\top H(\theta^*)^{-1} G(\theta^*) H(\theta^*)^{-1} \nabla_\theta s_\theta(\mathbf{x}, t)|_{\theta^*}). \tag{103}$$

From there, we compute the squared error in the scores. We now specify the remainder term in the asymptotic $N \to \infty$ analysis: it is in $o(N)$ and justified under the standard technical conditions of van der Vaart (2000, Th. 5.23). We write it in expectation with respect to the law of $\hat{\theta}$,

$$\mathbb{E}_{p(\hat{\theta})}[(s_{\hat{\theta}}(\mathbf{x}, t) - s_{\theta^*}(\mathbf{x}, t))^2] = \frac{1}{N} e(\boldsymbol{x}, t, \lambda^*, \lambda, p) + o(N^{-1}) \tag{104}$$

where

$$e(\boldsymbol{x}, t, \lambda^*, \lambda, p) = \text{trace}\big(H(\theta^*)^{-1} G(\theta^*) H(\theta^*)^{-1} \nabla_\theta s_\theta(\mathbf{x}, t)|_{\theta^*} \nabla_\theta s_\theta(\mathbf{x}, t)|_{\theta^*}^\top\big). \tag{105}$$

And then in expectation with respect to the law of $(\boldsymbol{x}, t)$

$$\mathbb{E}_{p_1(\boldsymbol{x}), p_K(t), p(\hat{\theta})}[(s_{\hat{\theta}}(\mathbf{x}, t) - s_{\theta^*}(\mathbf{x}, t))^2] = \frac{1}{N} e(\theta^*, \lambda, p) + o(N^{-1}) \tag{106}$$

where

$$e(\theta^*, \lambda, p) = \text{trace}\big(H(\theta^*)^{-1} G(\theta^*) H(\theta^*)^{-1} \mathbb{E}_{p_1(\boldsymbol{x}), p_K(t), p(\hat{\theta})}[\nabla_\theta s_\theta(\mathbf{x}, t)|_{\theta^*} \nabla_\theta s_\theta(\mathbf{x}, t)|_{\theta^*}^\top]\big) \tag{107}$$

**Specifying the matrices**    The following matrices were used above: we now recall their definition, using the same notation as in Bach (2024, Section 4.7).

$$G(\theta^*) = \mathbb{E}_{p(t), p(\mathbf{z}), p_t(\mathbf{x} \,|\, \mathbf{z})}[\nabla_\theta l_\theta(\mathbf{x}, \mathbf{z}, t)|_{\theta^*} \nabla_\theta l_\theta(\mathbf{x}, \mathbf{z}, t)|_{\theta^*}^\top] \tag{108}$$

$$H(\theta^*) = \mathbb{E}_{p(t), p(\mathbf{z}), p_t(\mathbf{x} \,|\, \mathbf{z})}[\nabla_\theta^2 l_\theta(\mathbf{x}, \mathbf{z}, t)|_{\theta^*}]. \tag{109}$$

**Case of CTSM**    We specify

$$\nabla_\theta l_\theta(\mathbf{x}, \mathbf{z}, t) = -2\lambda(t)\left(s(\mathbf{x}, \mathbf{z}, t) - s_\theta(\mathbf{x}, t)\right) \cdot \nabla_\theta s_\theta(\mathbf{x}, t), \tag{110}$$

$$\nabla_\theta^2 l_\theta(\mathbf{x}, \mathbf{z}, t) = 2\lambda(t) \cdot \nabla_\theta s_\theta(\mathbf{x}, t) \nabla_\theta s_\theta(\mathbf{x}, t)^\top - 2\lambda(t)\left(s(\mathbf{x}, \mathbf{z}, t) - s_\theta(\mathbf{x}, t)\right) \cdot \nabla_\theta^2 s_\theta(\mathbf{x}, t) \tag{111}$$

and evaluate them at $\theta^*$. To simplify notations, we write $w(\mathbf{x}, \mathbf{z}, t) = s(\mathbf{x}, \mathbf{z}, t) - s_{\theta^*}(x, t) = \partial_t \log p_t(\mathbf{x} \,|\, \mathbf{z}) - \partial_t \log p_t(\mathbf{x})$.

$$\nabla_\theta l_\theta(\mathbf{x}, \mathbf{z}, t)|_{\theta^*} = -2\lambda(t) w(\mathbf{x}, \mathbf{z}, t) \cdot \nabla_\theta s_\theta(\mathbf{x}, t)|_{\theta^*}, \tag{112}$$

$$\nabla_\theta^2 l_\theta(\mathbf{x}, \mathbf{z}, t)|_{\theta^*} = 2\lambda(t) \nabla_\theta s_\theta(\mathbf{x}, t)|_{\theta^*} \nabla_\theta s_\theta(\mathbf{x}, t)|_{\theta^*}^\top - 2\lambda(t) w(\mathbf{x}, \mathbf{z}, t) \cdot \nabla_\theta^2 s_\theta(\mathbf{x}, t)|_{\theta^*}. \tag{113}$$

Finally, this yields

$$G(\theta^*) = 4\mathbb{E}_{p(t),p(\mathbf{z}),p_t(\mathbf{x}\,|\,\mathbf{z})} \left[ \lambda(t)^2 w(\mathbf{x},\mathbf{z},t)^2 \cdot \nabla_\theta s_\theta(\mathbf{x},t)|_{\theta^*} \nabla_\theta s_\theta(\mathbf{x},t)|_{\theta^*}^\top \right], \tag{114}$$

$$H(\theta^*) = 2\mathbb{E}_{p(t),p(\mathbf{z}),p_t(\mathbf{x}\,|\,\mathbf{z})} \left[ \lambda(t) \cdot \nabla_\theta s_\theta(\mathbf{x},t)|_{\theta^*} \nabla_\theta s_\theta(\mathbf{x},t)|_{\theta^*}^\top - \lambda(t) \cdot w(\mathbf{x},\mathbf{z},t) \cdot \nabla_\theta^2 s_\theta(\mathbf{x},t)|_{\theta^*} \right]. \tag{115}$$

$\square$

A sufficient condition to make the error null in Eq. 107, is to have $w(\mathbf{x},\mathbf{z},t) = 0$.

**Case of CTSM-v**   The derivations are largely the same. We have

$$l_\theta(\mathbf{x},\mathbf{z},t) = \lambda(t)\|\mathrm{vec}(s(\mathbf{x},\mathbf{z},t)) - \mathrm{vec}(s_\theta(\mathbf{x},t))\|^2 = \lambda(t) \sum_i \left( (s(\mathbf{x},\mathbf{z},t))_i - s_\theta(\mathbf{x},t)_i \right)^2, \tag{116}$$

where $\mathrm{vec}(s(\mathbf{x},\mathbf{z},t))_i := \partial_t \log p_t(x^i\,|\,\mathbf{x}^{<i},\mathbf{z})$ indicates the $i$-th component of the vector $\mathrm{vec}(s(\mathbf{x},\mathbf{z},t)) = [\partial_t \log p_t(x^i\,|\,\mathbf{x}^{<i},\mathbf{z})]_{i\in[\![1,D]\!]}^\top$.

All that remains to specify the error are the matrices $\mathbf{G}$ and $\mathbf{H}$. We have

$$\nabla_\theta l_\theta(\mathbf{x},\mathbf{z},t) = -2\lambda(t) \sum_i (s(\mathbf{x},\mathbf{z},t)_i - s_\theta(\mathbf{x},t)_i)\, \nabla_\theta s_\theta(\mathbf{x},t)_i, \tag{117}$$

$$\nabla_\theta^2 l_\theta(\mathbf{x},\mathbf{z},t) = 2\lambda(t) \sum_i \nabla_\theta s_\theta(\mathbf{x},t)_i \nabla_\theta s_\theta(\mathbf{x},t)_i^\top - 2\lambda(t) \sum_i (s(\mathbf{x},\mathbf{z},t)_i - s_\theta(\mathbf{x},t)_i)\, \nabla_\theta^2 s_\theta(\mathbf{x},t)_i. \tag{118}$$

We now wish to evaluate these at $\theta^*$. To simplify notations, we now denote by $w(\mathbf{x},\mathbf{z},t)_i = s(\mathbf{x},\mathbf{z},t)_i - s_{\theta^*}(\mathbf{x},t)_i = \partial_t \log p_t(x^i\,|\,\mathbf{x}^{<i},\mathbf{z}) - \mathbb{E}_{p_t(\mathbf{z}\,|\,\mathbf{x})}\left[\partial_t \log p_t(x^i\,|\,\mathbf{x}^{<i},\mathbf{z})\right]$. Now we can write

$$\nabla_\theta l_\theta(\mathbf{x},\mathbf{z},t) = -2\lambda(t) \sum_i w(\mathbf{x},\mathbf{z},t)_i \nabla_\theta s_\theta(\mathbf{x},t)_i|_{\theta^*}, \tag{119}$$

$$\nabla_\theta^2 l_\theta(\mathbf{x},\mathbf{z},t) = 2\lambda(t) \sum_i \nabla_\theta s_\theta(\mathbf{x},t)_i|_{\theta^*} \nabla_\theta s_\theta(\mathbf{x},t)_i|_{\theta^*}^\top - 2\lambda(t) \sum_i w(\mathbf{x},\mathbf{z},t)_i \nabla_\theta^2 s_\theta(\mathbf{x},t)|_{\theta^*}. \tag{120}$$

As a result, we have

$$G(\theta^*) = 4\mathbb{E}_{p(t),p(\mathbf{z}),p_t(\mathbf{x}\,|\,\mathbf{z})} \left[ \lambda(t)^2 \left( \sum_i w(\mathbf{x},\mathbf{z},t)_i \nabla_\theta s_\theta(\mathbf{x},t)_i|_{\theta^*} \right)^2 \right], \tag{121}$$

$$H(\theta^*) = 2\mathbb{E}_{p(t),p(\mathbf{z}),p_t(\mathbf{x}\,|\,\mathbf{z})} \left[ \lambda(t) \sum_i \nabla_\theta s_\theta(\mathbf{x},t)_i|_{\theta^*} \nabla_\theta s_\theta(\mathbf{x},t)_i|_{\theta^*}^\top - \lambda(t) \sum_i w(\mathbf{x},\mathbf{z},t)_i \nabla_\theta^2 s_\theta(\mathbf{x},t)|_{\theta^*} \right]. \tag{122}$$

A sufficient condition to make the error null in Eq. 107, is to have $w(\mathbf{x},\mathbf{z},t)_i = 0$ for all $i$.

Table 2: Results on Gaussians with $D$ being 2, 5 or 10. $D$ is dimensionality, MSE is MSE to ground truth reported in the form of [mean, std], T is average time per step in ms. Unif indicates uniform weighting, Stein indicates Stein score normalization and Time indicates time score normalization, with Time 0 indicating using the real $c$ and Time 1 indicating using $c = 1$.

| | $D = 2$ | | $D = 5$ | | $D = 10$ | |
| --- | --- | --- | --- | --- | --- | --- |
| Algo | MSE | T | MSE | T | MSE | T |
| TSM+Unif | [0.21, 0.036] | 11.1 | [2.982, 1.738] | 12.5 | [12.478, 6.026] | 12.1 |
| TSM+Stein | [0.253, 0.142] | 13.4 | [2.408, 1.205] | 15.4 | [7.343, 1.378] | 14.0 |
| CTSM+Time 0 | [0.158, 0.049] | **3.9** | [1.37, 0.821] | 6.3 | [16.285, 11.047] | 5.3 |
| CTSM+Time 1 | [**0.078**, 0.017] | 4.5 | [0.987, 0.28] | 6.0 | [10.032, 5.476] | **4.8** |
| CTSM-v+Time 0 | [0.175, 0.045] | 8.3 | [0.86, 0.199] | 5.2 | [4.331, 0.727] | 5.1 |
| CTSM-v+Time 1 | [0.104, 0.014] | 4.0 | [**0.814**, 0.219] | **5.0** | [**1.616**, 0.203] | 4.9 |

Table 3: Results on Gaussians with $D$ being 15 or 20. $D$ is dimensionality, MSE is MSE to ground truth reported in the form of [mean, std], T is average time per step in ms. Unif indicates uniform weighting, Stein indicates Stein score normalization and Time indicates time score normalization, with Time 0 indicating using the real $c$ and Time 1 indicating using $c = 1$.

| | $D = 15$ | | $D = 20$ | |
| --- | --- | --- | --- | --- |
| Algo | MSE | T | MSE | T |
| TSM+Unif | [74.932, 60.02] | 13.8 | [335.45, 83.226] | 13.0 |
| TSM+Stein | [91.328, 48.905] | 14.3 | [329.779, 156.634] | 12.9 |
| CTSM+Time 0 | [36.922, 20.238] | 6.2 | [125.234, 30.715] | **3.9** |
| CTSM+Time 1 | [61.902, 19.891] | 5.5 | [50.756, 12.708] | 4.7 |
| CTSM-v+Time 0 | [16.529, 3.101] | 5.4 | [**41.945**, 13.973] | 5.0 |
| CTSM-v+Time 1 | [**8.88**, 1.921] | **4.8** | [43.861, 17.132] | 5.9 |

# E. Additional Experimental Results

**Distributions with high discrepancies**  We report the results of the algorithms under different settings and different weighting schemes. For TSM we additionally report the results under uniform weighting, i.e. $\lambda(t) = 1$.

**Gaussians**  We report the main results in Table 2 and Table 3. CTSM-v is consistently among the fastest and the best. The plot in the main paper is generated using TSM with Stein score normalization, CTSM with time score normalization and $c = 1$ and CTSM with time score normalization and $c = 1$.

We additionally report the results of using time score normalization for TSM in Table 4. We did not observe decisive improvements, and remark that CTSM-v yields better results with the same weighting scheme.

Table 4: Additional results on Gaussians. $D$ is dimensionality, MSE is MSE to ground truth reported in the form of [mean, std], T is average time per step in ms. Unif indicates uniform weighting, Stein indicates Stein score normalization and Time indicates time score normalization, with Time 0 indicating using the real $c$ and Time 1 indicating using $c = 1$.

| | TSM+Time 0 | | TSM+Time 1 | |
| --- | --- | --- | --- | --- |
| D | MSE | T | MSE | T |
| 2 | [0.217, 0.063] | 11.7 | [0.451, 0.206] | 11.6 |
| 5 | [3.764, 2.107] | 13.1 | [5.088, 4.481] | 12.0 |
| 10 | [13.647, 2.953] | 13.9 | [30.196, 12.414] | 12.3 |
| 15 | [96.588, 53.982] | 14.5 | [99.062, 34.036] | 31.8 |
| 20 | [218.046, 70.411] | 14.2 | [135.942, 53.202] | 13.9 |

Table 5: Results on GMMs with $\sigma = 1.0$. $k$ determines the distance between two GMM components, MSE is MSE to ground truth reported in the form of [mean, std], T is average time per step in ms. Unif indicates uniform weighting, Stein indicates Stein score normalization and Time indicates time score normalization, with Time 0 indicating using the real $c$ and Time 1 indicating using $c = 1$.

| | k=0.5 | | k=1.0 | | k=2.0 | |
|---|---|---|---|---|---|---|
| Algo | MSE | T | MSE | T | MSE | T |
| TSM+Unif | [**173.473**, 52.466] | 28.9 | [276.545, 97.042] | 12.6 | [14643.815, 13997.568] | 61.8 |
| TSM+Stein | [232.948, 133.647] | 14.6 | [459.645, 260.768] | 17.6 | [3427.258, 3545.452] | 12.0 |
| CTSM+Time 0 | [880.47, 172.594] | **4.1** | [480.847, 151.097] | 4.5 | [646.945, 210.44] | 4.7 |
| CTSM+Time 1 | [923.082, 131.758] | 4.6 | [460.546, 186.5] | 4.2 | [547.603, 200.504] | 4.8 |
| CTSM-v+Time 0 | [173.804, 108.326] | 4.8 | [211.046, 69.472] | **4.0** | [319.981, 100.91] | 7.4 |
| CTSM-v+Time 1 | [221.519, 98.112] | 5.8 | [**181.082**, 68.879] | 4.7 | [**266.486**, 150.877] | **4.3** |

Table 6: Results on GMMs with $\sigma = \sqrt{2.0}$. $k$ determines the distance between two GMM components, MSE is MSE to ground truth reported in the form of [mean, std], T is average time per step in ms. Unif indicates uniform weighting, Stein indicates Stein score normalization and Time indicates time score normalization, with Time 0 indicating using the real $c$ and Time 1 indicating using $c = 1$.

| | k=0.5 | | k=1.0 | | k=2.0 | |
|---|---|---|---|---|---|---|
| Algo | MSE | T | MSE | T | MSE | T |
| TSM+Unif | [1106.178, 550.442] | 33.3 | [1293.421, 270.072] | 12.5 | [6614.483, 1169.068] | 13.5 |
| TSM+Stein | [1460.023, 502.921] | 39.0 | [1564.266, 360.361] | 36.0 | [5180.453, 1786.018] | 12.7 |
| CTSM+Time 0 | [1934.401, 269.515] | 4.5 | [1872.342, 467.047] | 4.6 | [5961.52, 683.578] | 4.6 |
| CTSM+Time 1 | [2113.975, 403.59] | 8.0 | [2238.267, 123.69] | 4.6 | [6017.094, 344.537] | 4.0 |
| CTSM-v+Time 0 | [745.15, 158.202] | 4.6 | [1558.495, 379.161] | 4.5 | [5009.627, 1943.244] | 8.5 |
| CTSM-v+Time 1 | [762.231, 288.029] | 5.3 | [1762.826, 431.333] | 4.8 | [9226.993, 861.191] | 9.2 |

**Gaussian mixtures** We report the main results on Gaussian mixtures in Table 5. We set $\sigma$ in the Schrödinger bridge probability path to $1.0$ due to strong empirical results while enabling direct comparisons between TSM and CTSM(-v).

We additionally report the results with $\sigma = \sqrt{2.0}$ in Table 6 and the results with $\sigma = 0.0$ in Table 7. We observe that, setting $\sigma = \sqrt{2.0}$ results in worse performances for all methods. For TSM under uniform weighting, one can consider using $\sigma = 0.0$, in which case the performance improves, though CTSM-v under $\sigma = 1.0$ remains competitive.

Table 7: Results on GMMs with $\sigma = 0.0$. $k$ determines the distance between two GMM components, MSE is MSE to ground truth reported in the form of [mean, std], T is average time per step in ms. Unif indicates uniform weighting, Stein indicates Stein score normalization and Time indicates time score normalization, with Time 0 indicating using the real $c$ and Time 1 indicating using $c = 1$.

| | k=0.5 | | k=1.0 | | k=2.0 | |
|---|---|---|---|---|---|---|
| Algo | MSE | T | MSE | T | MSE | T |
| TSM+Unif | [148.688, 97.058] | 14.6 | [70.908, 5.85] | 12.9 | [898.016, 847.255] | 49.1 |

# F. Experimental Details

## F.1. Bug of TSM Implementation for Toy Experiments in Choi et al. (2022)

We observed a bug for the TSM implementation of the code of Choi et al. (2022). Recall that the TSM objective is given by

$$
\begin{aligned}
\mathcal{L}_{\text{TSM}}(\theta) = 2\mathbb{E}_{p_0(\mathbf{x})}[s_\theta(\mathbf{x}, 0)] - 2\mathbb{E}_{p_1(\mathbf{x})}[s_\theta(\mathbf{x}, 1)]+ \\
\mathbb{E}_{p(t,\mathbf{x})}[2\partial_t s_\theta(\mathbf{x}, t) + 2\dot{\lambda}(t)s_\theta(\mathbf{x}, t) + \lambda(t)s_\theta(\mathbf{x}, t)^2].
\end{aligned}
\tag{123}
$$

However, Choi et al. (2022) implemented

$$
\begin{aligned}
\mathcal{L}_{\text{TSM}}(\theta) = 2\mathbb{E}_{p_0(\mathbf{x})}[s_\theta(\mathbf{x}, 0)] - 2\mathbb{E}_{p_1(\mathbf{x})}[s_\theta(\mathbf{x}, 1)]+ \\
\mathbb{E}_{p(t,\mathbf{x})}[2\partial_t s_\theta(\mathbf{x}, t) + \dot{\lambda}(t)s_\theta(\mathbf{x}, t) + \lambda(t)s_\theta(\mathbf{x}, t)^2],
\end{aligned}
\tag{124}
$$

i.e. the scaling in front of $\dot{\lambda}(t)s_\theta(\mathbf{x}, t)$ is incorrect. We remark that this bug only applies when attempting to train purely based on TSM objective on toy experiments.

## F.2. Implementation Details

Our implementation of TSM is largely based on the code provided by Choi et al. (2022). However, especially for other than the EBM experiments, we improve their code in several ways. Apart from bug fixes, we use analytical expressions for the weighting quantities.

For both TSM and CTSM, following Choi et al. (2022), we add a small number $\epsilon$ to the time during training and inference. We follow the convention that $\epsilon$ is added when the probability path results in approximately degenerate distribution at that time. For the toy experiments, we set $\epsilon = 1e - 5$, while for EBM experiments we set $\epsilon = 1e - 4$ during training and $\epsilon = 1e - 5$ during inference.

For experiments apart from EBM, for each task we employ a fixed validation set of size 10000 and select the learning rates based on results on the sets. After a certain number of steps, an evaluation step is performed, and the model is evaluated based on both the validation set and a test set, consisting of 10000 samples dynamically generated based on the data generation process. The best test set results are obtained by selecting the steps corresponding to the best validation set results.

Following Choi et al. (2022), the density ratios are evaluated using the initial value problem ODE solver as implemented in SciPy (Virtanen et al., 2020), where we use the default RK45 integrator (Dormand & Prince, 1980) with $rtol = 1e - 6$ and $atol = 1e - 6$.

## F.3. Distributions with High Discrepancies

The experimental setup is similar to Choi et al. (2022). We use as score model a simple MLP with structure $[D + 1, 256, 256, 256, N_{output}]$ and ELU activation (Clevert et al., 2016) based on Choi et al. (2022), where $D$ is the dimensionality of the data and $N_{output} = D$ for CTSM-v and 1 otherwise. Note that the input shape is $D + 1$, as the time $t$ is concatenated to the input. All models are trained for 20000 iterations. After each 1000 iterations, the model is evaluated. For each scenario, the best learning rate is selected based on the best val set performances of a single run. Afterwards two runs under the same learning rate but different random seeds are run, and the final results on the test set is reported.

**Gaussians**    Following Choi et al. (2022), we employ the variance-preserving probability path, with $\alpha_t = t$.

The learning rate is tuned between $[5e - 4, 1e - 3, 2e - 3, 5e - 3, 1e - 2]$. Following Choi et al. (2022), the MSEs are evaluated using samples from both $p_0$ and $p_1$.

**Gaussian mixtures**    The learning rate is tuned between $[1e - 4, 2e - 4, 5e - 4, 1e - 3, 2e - 3, 2e - 3, 5e - 3, 1e - 2, 2e - 2, 5e - 2]$. There is one case where the selected learning rate for each algorithm is the smallest, and we manually verify that using lrs $5e - 5$ or $2e - 5$ does not result in improved results. Following Choi et al. (2022), the MSEs are evaluated using samples from both $p_0$ and $p_1$.

The two components are isotropic, with the covariance given by $\sigma^2 \mathbf{I}$. We use k to specify the distance between the means of the two components as a multiple of the standard deviation $\sigma$.

We know that the mean of a GMM is simply given by the mean of the means of each component, while the covariance of a GMM with two components of equal weights is given by the following formula

$$\mathbf{\Sigma} = \frac{1}{2}\,\mathbf{\Sigma}_1 + \frac{1}{2}\,\mathbf{\Sigma}_2 + \frac{1}{4}(\boldsymbol{\mu}_1 - \boldsymbol{\mu}_2)(\boldsymbol{\mu}_1 - \boldsymbol{\mu}_2)^\top. \tag{125}$$

Consider the case where $\boldsymbol{\mu}_1 - \boldsymbol{\mu}_2 = \mathrm{k}\sigma$. One has that, in order for the GMM to have variance equal to 1 in each dimension, $\sigma = \sqrt{4/(4 + \mathrm{k}^2)}$. The means of the two components are given by $\boldsymbol{\mu} - \frac{1}{2}\mathrm{k}\sigma$ and $\boldsymbol{\mu} + \frac{1}{2}\mathrm{k}\sigma$, respectively.

In principle, using $var = 2$ for SB path results in preserved variance along the path. However, empirically we observe that it is beneficial to use a smaller variance, e.g. $var = 1$.

## F.4. Mutual Information Estimation

The probability path is given by

$$p_t(\mathbf{x} \mid \mathbf{z}) = \mathcal{N}(\mathbf{x} \mid t\,\mathbf{x}_1, (1 - t^2)\,\mathbf{I}). \tag{126}$$

The derivations for the objective of TSM objective can be found in Choi et al. (2022). Here we derive the training objective for the CTSM-v objective.

Using similar settings and notations as in Choi et al. (2022), we parameterize a single matrix $\mathbf{S}$, as defined below.

Denote the covariance matrix of $p_1$ as $\mathbf{\Sigma}$. Use $\mathbf{S}$ to denote $\mathbf{\Sigma} - \mathbf{I}$.

Recall that the true time score is given by the posterior expectation of $\partial_t \log p_t(\mathbf{x} \mid \mathbf{z})$. We have

$$\log p(\mathbf{z}) = \log \mathcal{N}(\mathbf{z} \mid \mathbf{0}, \mathbf{\Sigma}) = -\frac{1}{2}\,\mathbf{z}^\top \mathbf{\Sigma}^{-1}\,\mathbf{z} + \text{const.}, \tag{127}$$

$$\log p_t(\mathbf{x} \mid \mathbf{z}) = \log \mathcal{N}(\mathbf{x} \mid t\,\mathbf{z}, (1 - t^2)\,\mathbf{I}) = -\frac{1}{2}t\,\mathbf{z}^\top \frac{1}{1 - t^2}t\,\mathbf{z} + \text{const.}. \tag{128}$$

The posterior distribution $p_t(\mathbf{z} \mid \mathbf{x})$ can be solved in closed-form, which is a Gaussian distribution, with covariance $\bar{\mathbf{\Sigma}} = \left(\mathbf{\Sigma}^{-1} + \frac{t^2}{1-t^2}\,\mathbf{I}\right)^{-1}$ and mean $\frac{t}{1-t^2}\bar{\mathbf{\Sigma}}\,\mathbf{x}$. Similar to Choi et al. (2022), the above quantities can be expressed in terms of the inverse of $\mathbf{I} + t^2\,(\mathbf{\Sigma} - \mathbf{I}) = (1 - t^2)\left(\mathbf{I} + \frac{t^2}{1-t^2}\,\mathbf{\Sigma}\right)$; we have

$$\left(\mathbf{\Sigma}^{-1} + \frac{t^2}{1 - t^2}\,\mathbf{I}\right)^{-1} = \left(\mathbf{\Sigma}^{-1}\left(\mathbf{I} + \frac{t^2}{1 - t^2}\,\mathbf{\Sigma}\right)\right)^{-1} \tag{129}$$

$$= \left(\mathbf{I} + \frac{t^2}{1 - t^2}\,\mathbf{\Sigma}\right)^{-1}\mathbf{\Sigma} = (1 - t^2)\left(\mathbf{I} + t^2\,(\mathbf{\Sigma} - \mathbf{I})\right)^{-1}\mathbf{\Sigma}. \tag{130}$$

The expectation of $\partial_t \log p_t(\mathbf{x} \mid \mathbf{z})$, which by definition is also the value of $\partial_t \log p_t(\mathbf{x})$, can also be obtained in closed-form. The expectation of the individual entries of $\partial_t \log p_t(\mathbf{x} \mid \mathbf{z})$ are also given in closed-form.

$$[\partial_t \log p_t(\mathbf{x} \mid \mathbf{z})]_i = \frac{t}{1 - t^2} - \frac{t}{(1 - t^2)^2}\left[(\mathbf{x} - t\,\mathbf{x}_1)^2\right]_i + \frac{1}{1 - t^2}\left[(\mathbf{x} - t\,\mathbf{x}_1)\,\mathbf{x}_1\right]_i, \tag{131}$$

$$\mathbb{E}_{p_t(\mathbf{z} \mid \mathbf{x})}\left[\partial_t \log p_t(\mathbf{x} \mid \mathbf{z})\right]_i = \frac{t(1 - t^2) - t\left(\|\bar{\boldsymbol{\mu}}_i\|^2 + \bar{\mathbf{\Sigma}}_{ii}\right) - t\|\mathbf{x}_i\|^2 + (t^2 + 1)\,\mathbf{x}_i\,\bar{\boldsymbol{\mu}}_i}{(1 - t^2)^2}, \tag{132}$$

where $\bar{\boldsymbol{\mu}}$ and $\bar{\mathbf{\Sigma}}$ are the mean and covariance of the posterior distribution as discussed above. As such, perhaps unsurprisingly, CTSM-v does not induce much computational overhead above TSM.

For CTSM objective, the model is trained to match the time score, while for CTSM-v objective, the model is trained to match the entire vec $(\partial_t \log p_t(\mathbf{x} \mid \mathbf{z}))$.

The hyperparameters are inspired by Choi et al. (2022) and listed in Table 8. For all methods, the learning rates are tuned between $1e - 4$, $1e - 3$ and $1e - 2$.

Table 8: Hyperparameters for MI experiment. After every eval freq steps, an evaluation is performed, with the first result after the first eval freq steps.

| D | n iters | eval freq | batch size |
|---|---|---|---|
| 40 | 20001 | 2000 | 512 |
| 80 | 50001 | 5000 | 512 |
| 160 | 200001 | 5000 | 512 |
| 320 | 400001 | 8000 | 256 |

## F.5. Energy-based Modeling

### F.5.1. GENERAL METHODOLOGY

We employ the same variance-preserving probability path as used in Choi et al. (2022), which in turn comes from diffusion models literature (Ho et al., 2020; Song et al., 2021b).

For reproducing TSM results, we use a batch size of $500$ and use polynomial interpolation with buffer size $100$, matching the reported hyperparameters in Choi et al. (2022). Following Choi et al. (2022), we tune the step size of TSM between $[2e-4, 5e-4, 1e-3]$. For CTSM-v, we largely reuse the hyperparameters, while tuning the step size between $[5e-4, 1e-3, 2e-3]$.

For CTSM-v objective, we parameterize the model to output the time score normalized by the approximate variance. Specifically, for a given $t$, we calculate $\mathrm{Var}\left(\partial_t \log p_t(\mathbf{x} \,|\, \mathbf{z})\right)$ where $c$ is assumed to be $1$, and the score network is trained to predict $\frac{\partial_t \log p_t(\mathbf{x})}{\mathrm{Std}(\partial_t \log p_t(\mathbf{x} \,|\, \mathbf{z}))}$; this ensures that the regression target is zero mean and having reasonable variances across $t$.

In previous works (Rhodes et al., 2020; Choi et al., 2022), different normalizing flows are fitted to the data, and DRE is carried out making use of the flows.

The flows can naturally be utilized in different ways. Denote the latent space of the flow as $\mathbf{u}$, and the ambient space of the flow as $\mathbf{x}$. Choi et al. (2022) consider the following scheme:

1. An SDE is defined on $\mathbf{u}$ space, interpolating between Gaussian and the empirical distribution induced by final samples on $\mathbf{u}$ space obtained by transforming the data points from $\mathbf{x}$ space,

2. Intermediate samples on $\mathbf{u}$ space are transformed into $\mathbf{x}$ space using the flow, inducing a time varying distribution on $\mathbf{x}$ space,

3. The score network takes as input $\mathbf{x}$ and $t$, and is trained to predict the time score.

Note that a flow is a bijection. Consider a time-varying density $p_t(\mathbf{x})$. For any $t$, we use the same bijective transformation $T$ to obtain the pair of $\mathbf{u}$ and $\mathbf{x}$. We have

$$\partial_t \log p_t(\mathbf{x}) = \partial_t \log \left( p_t(\mathbf{u}) \left| \det J_T(\mathbf{u}) \right|^{-1} \right) = \partial_t \log p_t(\mathbf{u}) + \partial_t \log \left| \det J_T(\mathbf{u}) \right|^{-1} = \partial_t \log p_t(\mathbf{u}). \quad (133)$$

As such, the time score is invariant across bijections.

With CTSM, inspired by previous approaches, we also consider a probability path in $\mathbf{u}$ space. One needs the time score of the conditional distribution, which needs to be computed in $\mathbf{u}$ space. One can in principle train the score network either by feeding in coordinates of points in the $\mathbf{u}$ space or the corresponding coordinates in $\mathbf{x}$ space, where the conditional target vector field is computed in $\mathbf{u}$ space.

1. An probability path is defined on $\mathbf{u}$ space, interpolating between Gaussian and the empirical distribution induced by samples,

2. The score network takes as input either $\mathbf{x}$ or $\mathbf{u}$ along with $t$, and learns the time score.

Note that it is correct to feed in the score network either $\mathbf{x}$ or $\mathbf{u}$; when the model takes as input $\mathbf{x}$, one can interpret that the normalizing flows is a part of the score network, i.e. $\tilde{\mathbf{s}}_\theta(\mathbf{u}, t) = \mathbf{s}_\theta(\mathbf{f}^{-1}(\mathbf{x}), t)$, where $\mathbf{f}$ is the normalizing flows that is

fixed and does not need to be learned and $\mathbf{s}$ is the score network that we parameterize. As such, the correctness is guaranteed by standard CTSM / CTSM-v identities. We empirically observe that directly feeding in $\mathbf{x}$ coordinates leads to better BPD estimates.

We remark that both TSM and CTSM need to map between $\mathbf{u}$ and $\mathbf{x}$ coordinates using the normalizing flows. However, while CTSM only need to map both $\mathbf{u}$ to $\mathbf{x}$ and $\mathbf{x}$ to $\mathbf{u}$ exactly once, TSM requires an extra $\mathbf{u}$ to $\mathbf{x}$ map due to needed by the boundary condition.

### F.5.2. Experimental Details

In terms of EBM with Gaussian flows, we observe that, possibly due to the parameterization, models trained using CTSM-v may require a larger number of integration steps compared with TSM when evaluating the density ratio using an ODE integrator with specific error tolerances as described in Section F.2: on MNIST test set with batch size 1000, TSM requires on average $489.2$ evaluations, while CTSM-v requires on average $830.6$ evaluations. However, we remark that it is unclear what the true time scores are like.

Especially with TSM, the BPD estimate on the validation set can vary greatly. In general, we report the final results using the checkpoint that resulted in the best validation set result, unless the BPD estimate falls below $0$, in which case we consider the training as unstable.

Previous works (Rhodes et al., 2020; Choi et al., 2022) experimented with performing DRE utilizing copula flows and RQ-NSF flows. We observe that CTSM-v suffers from unstable optimization, with the BPD dropping at first but shoots back to a large value afterwards. We hypothesize that the parameterization does not agree with the inductive bias of the employed score network.

When training CTSM-v in the ambient space, we employ as score network a more advanced U-Net largely based on Song et al. (2021b) and Choi et al. (2022), which, among others, employs residual blocks. As noted by Choi et al. (2022), residual blocks result in unstable training for TSM. However, CTSM-v works well with this version of the U-Net. We also experimented with training using TSM with the U-Net as used by Choi et al. (2022) in the ambient space, but found the BPD estimates on the val set to be generally larger than $5$ and did not explore it further.

With Gaussian flows, all experiments were run using one NVIDIA V100 GPU each. With ambient space, the models were trained and evaluated using one NIVDIA A100 GPU each, while the running times were obtained based on 10000 steps using one NVIDIA V100 GPU each. We observed that the training dynamics of the models may vary across the employed GPUs, even with the other settings kept the same.

### F.6. Sampling

We employ annealed MCMC to draw samples from the learned score network. We draw a total of 100 samples. For each sample, we construct 1000 intermediate distributions, where each intermediate distribution is targeted using a single HMC step. The intermediate distributions are constructed by linearly interpolating between 0 and 1 and setting

$$\log p_t(\mathbf{x}) = \log p_0(\mathbf{x}) + \int_{\tau=0}^{t} \partial_\tau \log p_\tau(\mathbf{x})\mathrm{d}\tau. \tag{134}$$

After which, we run another 100 steps of HMC to further refine the samples.

Each step of HMC contains 10 leapfrog steps. When using Gaussian flows, we observe a correlation between sample quality and the estimated log constant, where the sample quality is good when the estimated log constant is close to $0$. Based on the observation, we tune the step size of HMC on a grid in the form of $[1e - n, 2.5e - n, 5e - n, 7.5e - n]$.

For Gaussian flows, we run annealed MCMC with a batch size of $100$. The first $64$ samples drawn from models trained using TSM and CTSM-v are shown in Figure 5.

For pixel space CTSM-v, we consider both annealed MCMC and the Probability Flow (PF) ODE (Song et al., 2021b). For annealed MCMC, we use a batch size of $50$ and run 2 individual batches. For PF ODE, we use a batch size of $64$. The first $64$ samples were reported in the main paper.

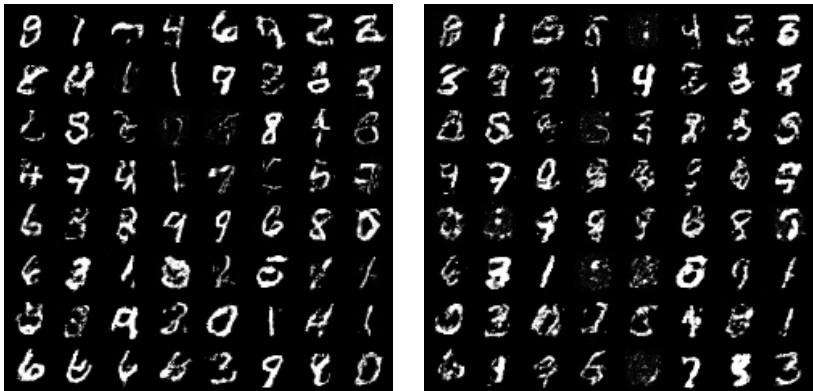

Figure 5: Left: samples drawn from a model trained with TSM, Gaussian flows; right: samples drawn from a model trained with CTSM-v, Gaussian flows.

