# OpenReview forum: "Density Ratio Estimation with Conditional Probability Paths"
_ICML.cc/2025/Conference — ICML 2025 poster_

### Official Review · Reviewer_B2JP · 2025-03-16

**Overall Recommendation:** 4

**Summary:**

This paper introduces a new method for density ratio estimation called conditional time score matching (CTSM). CTSM estimates the time score along a probability path connecting two densities instead of directly estimating the ratio of two densities. By conditioning on additional variables, the authors propose an easy-to-estimate objective and a faster variant. They further provide the theoretical guarantee of error bounds and then show strong empirical results, especially in high-dimensional settings.

## update after rebuttal:
I'm maintaining my score.

**Claims And Evidence:**

Yes, the claims are supported by clear theoretical analysis and empirical results.

**Essential References Not Discussed:**

The current set of references is robust.

**Experimental Designs Or Analyses:**

The experimental design in this paper is sound and consistent with prior works. However, in Table 1, the evaluation is limited with only a pretrained Gaussian normalizing flow. The paper could benefit from additional experiments from prior work. Specifically, the paper could include experiments using Copula interpolation and RQ-NSF interpolation following [Choi 2022] Table 1.


Choi 2022. Density Ratio Estimation via Infinitesimal Classification.

**Methods And Evaluation Criteria:**

The method and chosen benchmarks clearly evaluating the improvements over existing methods. The evaluation criteria follows prior work of Rhodes 2020 and Choi 2022 and they make sense for the problem.

**Other Comments Or Suggestions:**

Not applicable.

**Other Strengths And Weaknesses:**

See "Experimental Designs Or Analyses" for additional experiments.

**Questions For Authors:**

Not applicable.

**Relation To Broader Scientific Literature:**

This paper contributes to the broader research on density ratio estimation, which has numerous application in machine learning.

**Theoretical Claims:**

I review the key theoretical results and proofs (mainly section 5) in the paper. The proofs are correct and rigorous.

---

> ### Author Rebuttal · Authors · 2025-03-31
>
> Thank you for the very positive assessment. Below we respond to the comment on the MNIST experiment.
>
> We indeed showed results only for pre-trained Gaussian NF, since it was the most computationally stable. We have some experimental results also for Copula and RQ-NSF utilizing essentially the same score network as in [1] but found the optimization to be unstable.
>
> Moreover, after the submission we also experimented more with directly solving the same problem in the ambient space itself using a different score network, instead of using any pre-trained flow and its latent space. Whereas previous methods reported that using a pre-trained flow was important to obtain good results (see [1, 2]), our method works extremely well in the ambient space scenario, reaching a BPD of 1.03, outperforming any of the previous results utilizing a latent space. We also checked that the learnt time score in ambient space was capable of generating reasonable samples using two sampling methods: annealed MCMC and the Probability Flow ODE from [3]. We will also add this result.
>
> [1] Choi et al., Density Ratio Estimation via Infinitesimal Classification, AISTATS 2022
>
> [2] Rhodes et al., Telescoping Density-Ratio Estimation, NeurIPS 2020
>
> [3] Song et al., Score-Based Generative Modeling through Stochastic Differential Equations, ICLR 2021

---

### Official Review · Reviewer_h72Z · 2025-03-17

**Overall Recommendation:** 3

**Summary:**

The authors tackle the problem of estimating the ratio of two probability densities, improving upon speed and accuracy of prior work. They also establish a theorem on a guarantee of the error in the estimated density ratio. The method applies the "marginalization trick" (a la flow matching) to make the learning objective tractable. Essentially a latent variable is introduced to the stochastic interpolant between the test and target densities, and object modified to match the expectation of score matching marginalizing out thelatent. The distributions are chosen in a clever way similar to previous literature so that the objective is tractable (so called Conditional Time Score Matching objective). The authors also introduce a variant where a multivariate test and target distributions are broken up into n terms in an autoregressive fashion and the terms matched separately. A novel contribution is the design of a weighting function, time score normalization, to stabilize training. This could become a standard statistical method.

**Claims And Evidence:**

Claims in the paper are supported by ample ablation studies and other experiments (with uncertainty estimates).

**Essential References Not Discussed:**

N/A

**Experimental Designs Or Analyses:**

Experimental design is sound.

**Methods And Evaluation Criteria:**

Proposed evaluation makes sense for the problem at hand and are quite comprehensive.

**Other Comments Or Suggestions:**

N/A

**Other Strengths And Weaknesses:**

N/A

**Questions For Authors:**

I think the paper would benefit from some discussion on how this work isn't just a straightforward application of the ideas behind flow matching. Is this something you would consider?

**Relation To Broader Scientific Literature:**

Estimation of density ratios is a fundamental problem in generative modeling with many applications, as the paper describes.

**Theoretical Claims:**

I didn't review the proof for Theorem 4 and Proposition 5, which are in the supplementary materials.

---

> ### Author Rebuttal · Authors · 2025-03-31
>
> Thank you for your positive assessment. We agree that our learning objective (CTSM) is similar to the learning objective in flow matching (Eq 5, [1]). However, our method is not a straightforward application of flow matching given that, fundamentally, we learn a different quantity. Flow matching learns a velocity field while we learn a time score.
>
> Furthermore, our vectorized learning objective (CTSM-v) is a novel contribution of our work that is essential for high-dimensional experiments (see Fig 2). In fact, our CTSM-v objective has no obvious counterpart in the flow matching literature, to the best of our knowledge. Moreover, our time score matching weighting scheme is tailored for learning the time score and is different from weighting schemes used in flow matching (Eq 5, [1], i.e. a uniform weighting).
>
> We are happy to include these clarifications in the main text.
>
> [1] Lipman et al., Flow Matching for Generative Modeling, ICLR 2023

---

### Official Review · Reviewer_MEax · 2025-03-18

**Overall Recommendation:** 5

**Summary:**

This paper proposes the conditional time score matching (CTSM) objective, which is a variant of the time score matching (TSM) objective proposed in the recent density ratio estimation literature.
The CTSM objective provides a principled way to detour the computational drawback of TSM, which requires two automatic differentiation steps. The idea is to introduce a conditioning variable z, and the mathematical trick is essentially same as that of denoising score matching. The authors provide a vector variant that can be useful in practice.
They also analyze the quality of the estimated density $\hat{p}_1(x)$, where the integration of the time score is approximated by a discretization, assuming that $p_0(x)$ is given.
The experiments support that CTSM can perform as well as TSM with less computation.
Overall, it is a well-written paper with well-executed analyses and experiments.

## update after rebuttal

I thank the authors' response. I will keep my positive score.
For the final version, it'd be helpful if the authors further comment on the choice of KL divergence for the analysis and its limitation.

**Claims And Evidence:**

The theoretical claims are solid. Experimental validations are also thorough showing the benefit of CTSM compared to TSM.

**Essential References Not Discussed:**

References seem adequate.

**Experimental Designs Or Analyses:**

The experiments are solid enough to demonstrate the benefit.

**Methods And Evaluation Criteria:**

The proposed method is a very natural yet effective modification of TSM leading to computational efficiency.
The evaluation criteria in experiments seems also adequate.

**Other Comments Or Suggestions:**

N/A

**Other Strengths And Weaknesses:**

It was a pleasant read overall and I do not have any specific comment on weakness.

**Questions For Authors:**

- In theoretical guarantees, is there a particular reason for the KL divergence considered in the analysis? I am just curious if this is a convenient choice for the ease of analysis, or something else. Can this relationship be translated to, for example, the expected value of the square of the difference in the log density ratios?
- Is the last term a typo in Eq. (45)?

**Relation To Broader Scientific Literature:**

Density ratio estimation in general is a key task in machine learning.
Recent years density ratio estimation based on infinitesimal classification has received attention as a solution to the so-called "density chasm" problem. The paper's idea is not a rocket science, but it provides a very thorough analysis on the benefit of conditioning both in theory and practice.

**Theoretical Claims:**

I checked the proofs of the statements, but only skimming over the proof of Proposition 5.

---

> ### Author Rebuttal · Authors · 2025-03-31
>
> Thank you for the very positive assessment and for spotting the typo in Eq. 45 which is now corrected. Regarding our theoretical guarantees, we chose the KL divergence out of convenience for the derivations in Appendix D.3. A similar analysis could indeed be used to bound the expected value of the square of the difference in the log density ratios. To do so, we would start from Eq. 81 and replace the log densities with log density ratios.

---

### Decision · Program_Chairs · 2025-05-01

**Decision:**

Accept (poster)

**Comment:**

This paper proposes the conditional time score matching (CTSM) objective, which is a variant of the time score matching (TSM) objective proposed in the recent density ratio estimation literature. The CTSM objective provides a principled way to detour the computational drawback of TSM, which requires two automatic differentiation steps. The idea is to introduce a conditioning variable and apply the "marginalization trick" (a la flow matching) to make the learning objective tractable.

All reviewers are unanimously positive about this paper: the density ratio estimation problem is important, the methodology is clever and sound, and the paper is well-written with with well-executed analyses and experiments. Based on the reviews and my own reading, I happily recommend acceptance.